# An acute microglial metabolic response controls metabolism and improves memory

Anne Drougard[1,2]*, Eric H Ma[3], Vanessa Wegert[1,2], Ryan Sheldon[4], Ilaria Panzeri[1,2], Naman Vatsa[5], Stefanos Apostle[1], Luca Fagnocchi[1], Judith Schaf[2], Klaus Gossens[2], Josephine Völker[2], Shengru Pang[6], Anna Bremser[2], Erez Dror[2], Francesca Giacona[1], Sagar Sagar[2,7], Michael X Henderson[5], Marco Prinz[6,8,9], Russell G Jones[3], John Andrew Pospisilik[1,2]*

[1]Department of Epigenetics, Van Andel Research Institute, Grand Rapids, United States; [2]Max Planck Institute of Immunobiology and Epigenetics, Freiburg, Germany; [3]Department of Metabolism and Nutritional Programming, Van Andel Research Institute, Grand Rapids, United States; [4]Metabolomics and Bioenergetics Core, Van Andel Institute, Grand Rapids, United States; [5]Department of Neurodegenerative Sciences, Van Andel Research Institute, Grand Rapids, United States; [6]Institute of Neuropathology, Medical Faculty, University of Freiburg, Freiburg, Germany; [7]Department of Medicine II, University Hospital Freiburg, Freiburg, Germany; [8]Centre for NeuroModulation (NeuroModBasics), University of Freiburg, Freiburg, Germany; [9]Signaling Research Centers BIOSS and CIBSS, University of Freiburg, Freiburg, Germany

*For correspondence:
anne.drougard1@gmail.com (AD);
andrew.pospisilik@vai.org (JAndrewP)

## eLife Assessment

This **important** study demonstrates a link between an acute high fat diet, microglial metabolism and improved higher cognitive function. The evidence supporting the proposed mechanism in vivo is incomplete at this stage due to non-trivial technical limitations but the authors provide **convincing** in vitro metabolic characterization of primary microglia cultures to support the model. This work will be of interest to a broad audience in the field of neuroscience, metabolism, and immunology.

**Abstract** Chronic high-fat feeding triggers metabolic dysfunction including obesity, insulin resistance, and diabetes. How high-fat intake first triggers these pathophysiological states remains unknown. Here, we identify an acute microglial metabolic response that rapidly translates intake of high-fat diet (HFD) to a surprisingly beneficial effect on metabolism and spatial/learning memory. High-fat intake rapidly increases palmitate levels in cerebrospinal fluid and triggers a wave of microglial metabolic activation characterized by mitochondrial membrane activation and fission as well as metabolic skewing toward aerobic glycolysis. These effects are detectable throughout the brain and can be detected within as little as 12 hr of HFD exposure. In vivo, microglial ablation and conditional DRP1 deletion show that the microglial metabolic response is necessary for the acute effects of HFD. $^{13}$C-tracing experiments reveal that in addition to processing via β-oxidation, microglia shunt a substantial fraction of palmitate toward anaplerosis and re-release of bioenergetic carbons into the extracellular milieu in the form of lactate, glutamate, succinate, and intriguingly, the neuroprotective metabolite itaconate. Together, these data identify microglia as a critical nutrient regulatory node in the brain, metabolizing away harmful fatty acids and liberating the same carbons as alternate bioenergetic and protective

substrates for surrounding cells. The data identify a surprisingly beneficial effect of short-term HFD on learning and memory.

## Introduction

The central nervous system (CNS) plays a critical role in regulating glucose and energy homeostasis by sensing, integrating, and responding to peripheral signals (*Kleinridders et al., 2009*). Our understanding of CNS control over energy balance is mostly derived from investigation of distinct functional sets of hypothalamic neurons and their interactions with the periphery (*Grayson et al., 2013*). In addition, a growing body of evidence supports an important role for glial cells in regulating energy balance (*Argente-Arizón et al., 2017*). For example, astrocytes can regulate energy balance by processing glucose into lactate, which is then released to fuel and modulate surrounding neurons (*Pellerin et al., 1998*; *Magistretti and Allaman, 2018*; *Bélanger et al., 2011*; *Suzuki et al., 2011*). These latter findings demonstrate how metabolic intermediaries from the neuronal niche can directly influence CNS output.

Microglia are the third most abundant glial cell in the brain. Developmentally, they are derived from the yolk sac and are considered CNS immune cells. Like the peripheral immune compartment, microglial activation has been implicated in metabolic disease pathogenesis (*Thaler et al., 2012*; *Valdearcos et al., 2017*; *Kim et al., 2019*). Specifically, microglia in the mediobasal hypothalamus (MBH) become activated under conditions of chronic high-fat feeding in a process termed 'microgliosis'. High-fat diet (HFD)-induced activation has been associated with moderate increases in cytokine gene expression (*Thaler et al., 2012*; *Yi et al., 2017*) and pioneering studies testing both genetic manipulation of microglial NF-kB and microglial depletion with PLX5622 suggest that microglial activation is necessary for the metabolic impairments triggered by chronic HFD (*Valdearcos et al., 2017*). Recent evidence suggests a role for microglial mitochondrial alterations (fission) that can be detected within days of HFD administration (*Kim et al., 2019*). Microglia outside the hypothalamus also activate upon chronic HFD exposure (*Vinuesa et al., 2019*; *Hao et al., 2016*) and associate with cognitive impairment (*Spencer et al., 2017*; *Duffy et al., 2019*). A key outstanding question in the field, then, is how high-fat feeding first triggers microglial activation.

Here, we find that microglia respond to dietary fat within single 12 hr feeding cycle. By 3 days, HFD increases microglial mitochondrial membrane potential, inhibits complex II activity, and skews mitochondria toward fission. Using cerebrospinal fluid (CSF) metabolomics we demonstrate that microglia are directly exposed to increased palmitate and stearate levels within hours of the feeding switch, and using $^{13}$C-tracer studies, we find that microglia robustly metabolize fatty acids via β-oxidation. Interestingly, we find the same acute microglial metabolic response (aMMR) beyond the hypothalamus in vivo, in the hippocampus and cortex. aMMR is largely independent of transcriptional changes, and intriguingly, we find that it improves spatial and learning memory. Microglial depletion experiments and microglia-specific DRP deletion show that microglia and more specifically microglial mitochondrial fission are necessary for both the immediate whole-body glucose homeostatic response and the memory improvements to acute HFD. The data thus identify a non-transcriptional microglial metabolic mechanism that links detection of acute changes in dietary fat to changes in whole-body metabolism and memory.

## Results

### Acute HFD-induced metabolic changes are microglia dependent

To understand the nature of the immediate high-fat feeding response, we characterized the physiological effects of acute HFD (3d ad libitum feeding) in cohorts of C57BL/6J mice (*Figure 1A*). Metabolically healthy mice exhibit rapid metabolic shifts on this timescale, including fasting and post-absorptive (2 hr fast) hyperglycemia and an elevated insulin response to glucose (*Figure 1B–E*, *Figure 1—figure supplement 1A and B*; *Benani et al., 2012*; *Wang et al., 2001*). Glucose excursion, as revealed by baseline correction, and insulin tolerance were minimally impacted upon acute HFD (*Figure 1—figure supplement 1C*), an important contrast to chronic HFD (*Gregor and Hotamisligil, 2011*). These acute changes in glycemic control were transient, returning to homeostatic levels within 1 week of a return to normal chow ('Reverse Diet'; *Figure 1C–E*) and are not associated with

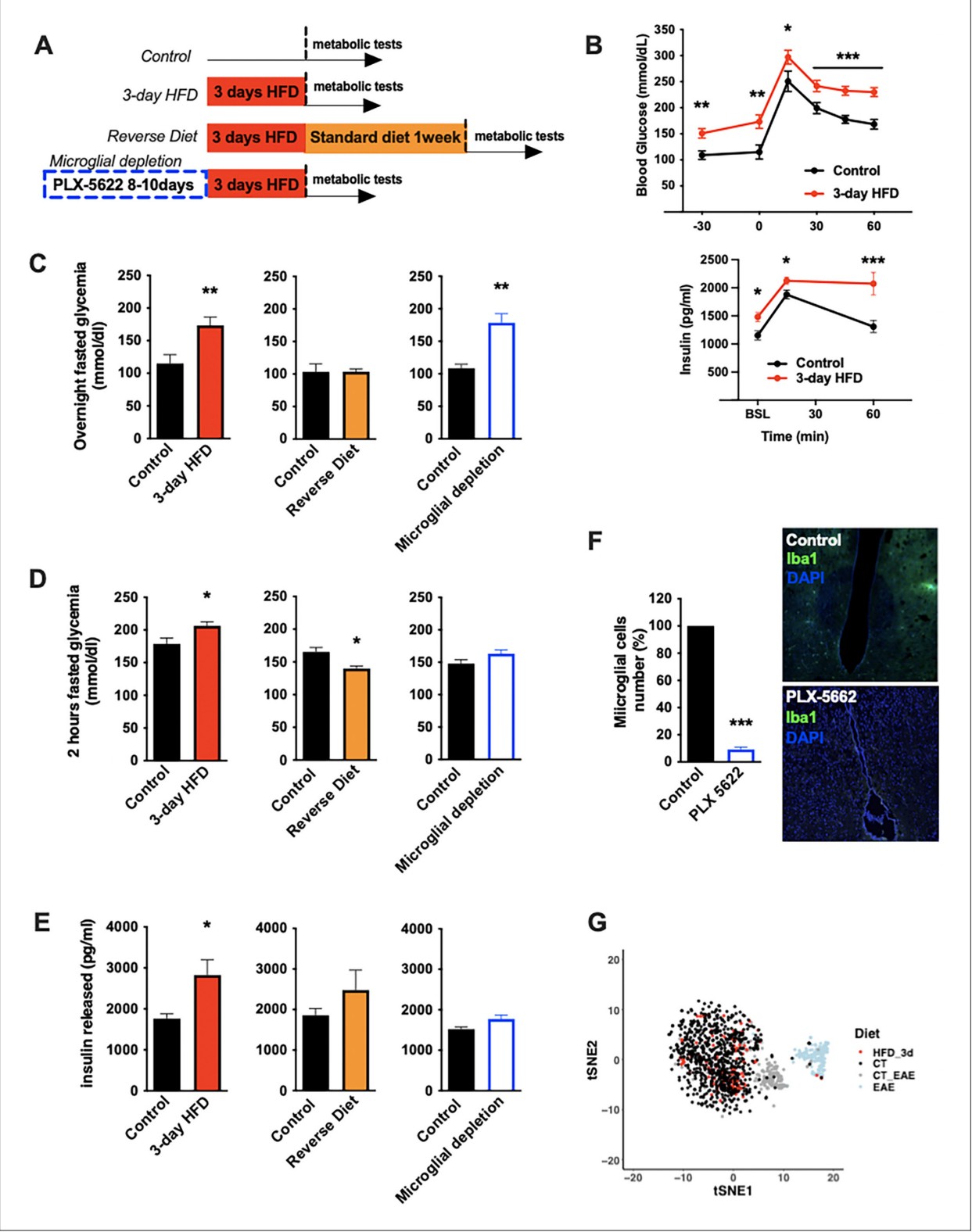

**Figure 1.** Acute HFD-induced metabolic changes are microglia dependent. (**A**) Schematic depicting the different treatments and diets followed by the mice groups. (**B**) Graphs showing the glucose tolerance test (OGTT) and the associated-insulin kinetic of C57Bl6/J male fed with control diet (Control) or fed with high-fat diet for 3 days (3-day HFD) (n=8). (**C**) Graph showing the overnight fasted glycemia from the mice groups depicting in (**A**). Schematic (n=5–11). (**D**) Graph showing the 2 hr fasted glycemia from the mice groups depicting in (**A**). Schematic (n=5–11). (**E**) Graph showing the insulin released after a glucose gavage from the mice groups depicting in (**A**). Schematic (n=5–11). (**F**) Microglial cells staining with Iba1 (green) in the brain slices from

*Figure 1 continued on next page*

*Figure 1 continued*

mice fed with 3 days HFD or mice depleted from their microglial cells with 1 week control diet complexed with PLX-5662 prior the 3 days HFD (PLX-5662) (n=5). (**G**) Single-cell RNA-sequencing (scRNAseq) data from hypothalamic microglial cells harvested from C57bl6/J male mice fed with control diet (CT) and high-fat diet for 3 days (HFD_3d) (n=5) merged with scRNAseq microglia dataset from mice presenting an experimental autoimmune encephalomyelitis (EAE). Data are presented as mean ± SEM. *p<0.05, **p<0.01, ***p<0.001 as determined by two-tailed Student's t-test and two-way ANOVA followed by Bonferroni post hoc test.

The online version of this article includes the following figure supplement(s) for figure 1:

**Figure supplement 1.** Acute HFD-induced metabolic changes are microglia dependent (supplement).

**Figure supplement 2.** Microglia-ablating drug PLX-5622 treatment doesn't modify body weight gain or food intake.

substantial body weight gain (*Figure 1—figure supplement 1D*). To test whether microglia contribute these effects, we treated mice with the microglia-ablating drug PLX-5622 (*Feng et al., 2017*) for 7–9 days and repeated the 3d HFD metabolic assessment. PLX-5622 treatment depleted hypothalamic microglia by >95% across animals (*Figure 1F*), without body weight gain or food intake modification (*Figure 1—figure supplement 2A and B*) and prevented induction of both the 3d HFD insulin hypersecretion and post-absorptive hyperglycemia (*Figure 1C–E*). Hyperglycemia associated with overnight fasting was unaffected by microglial depletion (*Figure 1C*). The latter indicates that acute HFD triggers microglia-dependent and independent metabolic effects. Thus, microglia are required for metabolic changes induced by the transition to high-fat feeding.

## A rapid microglial mitochondria response to HFD

Long-term exposure to HFD has previously been associated with hypothalamic microgliosis (increased number and activation of CD45$^{lo}$;CD11b$^+$ microglia) and monocyte infiltration (CD45$^{hi}$;Cd11b$^+$) (*Valdearcos et al., 2017*). Examination of 3d HFD responses in both wild-type animals and animals harboring a microglia-restricted eYFP lineage reporter showed no evidence of either increased hypothalamic microglial proliferation (*Figure 1—figure supplement 1E*) or monocyte infiltration (*Figure 1—figure supplement 1F*). Similarly, using the highly sensitive single-cell RNA-sequencing (scRNAseq) protocols, CEL-Seq2, we found no evidence of a significant microglial transcriptional response to this very short-term HFD exposure. The latter analysis included searches for changes in heterogeneity, skewing across sub-states, and appearance of new cell sub-states (*Figure 1G*, *Figure 1—figure supplement 1G*). The lack of response was especially clear when juxtaposed to experimental autoimmune encephalomyelitis (EAE)-triggered responses performed using the same purification and sequencing protocols in the same laboratory (EAE; *Figure 1G*). Thus, the central effects of acute HFD are distinct from those of chronic HFD, and have little to no effect on microglial expansion, transcription, heterogeneity, or inflammation in the hypothalamus.

A deep body of literature has highlighted how cellular metabolic changes are necessary and sufficient mediators of cell-type-specific function, work revitalised of late by the immunometabolism community (*Pearce et al., 2013*). We tested whether such changes in cellular metabolism might underpin the microglial response to acute high-fat feeding. Interestingly, mitochondrial membrane potential was increased in primary MBH microglia sorted from animals administered a 3d HFD (MitoTracker Deep Red; *Figure 2A and B*). Co-staining in the same samples showed no evidence of altered mitochondrial mass (MitoTracker Green; *Figure 2C*). This aMMR was detectable within 12 hr of high-fat feeding (*Figure 2B*). Importantly, aMMR appeared transient and distinct from chronic HFD response; parallel measures showed no induction of membrane potential in 4-week HFD animals (*Figure 2B*). Thus, microglia respond to acute HFD with a unique and specific acute mitochondrial metabolic response.

To understand the inputs that trigger aMMR, we performed targeted metabolomics of CSF (*Figure 2D*; *Table 1*). Overall, relatively few metabolites change upon 3d HFD. Nicotinamide-N-oxide was the only metabolite that decreased significantly; and six metabolites increased upon 3d HFD including methylmalonate, isovalerylcarnitine, nicotinic acid, glutarylcarnitine, and the fatty acids, palmitate (hexadecanoic acid) and stearate (octadecanoic acid). Palmitate and stearate, the only two metabolites that changed by >2-fold, are highly enriched in the HFD itself (Research Diets D12492) suggesting that microglia respond 'directly' to dietary fatty acid levels.

To test whether these observed fatty acid changes in the CSF might directly trigger aMMR, we switched to an in vitro primary neonatal microglia model and examined the effects of the more abundant of these, palmitate (*Li et al., 2018*; *Figure 2—figure supplement 1A and B*). Palmitate

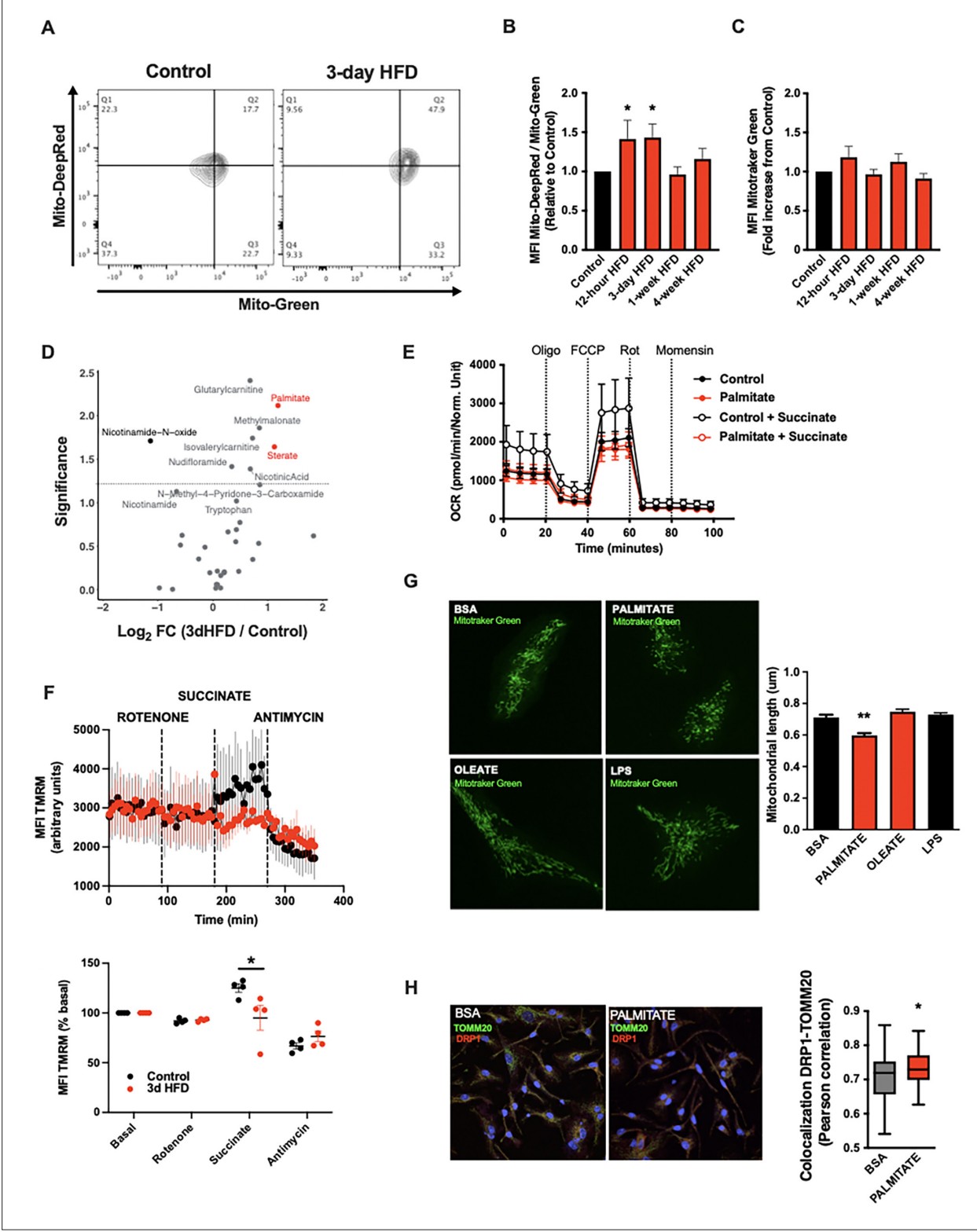

**Figure 2.** A rapid microglial mitochondria response to high-fat diet. (**A**) FACS plots depicting the ratio MitoTracker Deep Red/ MitoTracker Green from sorted microglial cells of C57Bl6/J male fed with a control diet (Control) or fed with high-fat diet for 3 days (3-day HFD). (**B**) Graph showing the ratio MitoTracker Deep Red/ MitoTracker Green from sorted hypothalamic microglial cells of C57Bl6/J male fed with a control diet (Control) or fed with high-fat diet for 12 hr, 3 days or 1–4 weeks (n=5–12). (**C**) Graph showing the MitoTracker Green fluorescence from sorted hypothalamic microglial cells of C57Bl6/J male fed with a control diet (Control) or fed with high-fat diet for 12 hr, 3 days or 1–4 weeks (n=5–12). (**D**) Volcano plot showing the

*Figure 2 continued on next page*

Figure 2 continued

metabolites content of cerebrospinal fluid from C57Bl6/J male fed with a control diet (Control) or fed with high-fat diet for 3 days (n=10). (E) Seahorse (±succinate added in the media during the experiment) on primary microglia challenged for 24 hr with BSA (control) or palmitate (experiment replicated three times). (F) Mitochondrial electron transport chain activity recorded with FACS after TMRM staining from sorted microglial cells of C57Bl6/J male fed with a control diet (Control) or fed with high-fat diet for 3 days (n=5). (G) Mitochondrial networks from primary microglia stained with MitoTracker Green after being challenged for 24 hr with BSA (control), palmitate, oleate, or LPS (n=40) and the mitochondrial length quantification graphs. (H) DRP1 colocalization with the mitochondrial network stained with TOMM20 on primary microglial cell after being challenged for 24 hr with BSA (control) and palmitate (n=40) and the colocalization quantification graphs. Data are presented as mean ± SEM. *p<0.05, **p<0.01, ***p<0.001 as determined by two-tailed Student's t-test and two-way ANOVA followed by Bonferroni post hoc test.

The online version of this article includes the following figure supplement(s) for figure 2:

**Figure supplement 1.** A rapid microglial mitochondria response to high-fat diet (supplement).

had no effect on either baseline or maximal oxygen consumption under standard culture conditions (*Figure 2E*). That said, when added in conjunction with succinate (to prime complex II-mediated OxPhos), palmitate-treated microglia were unable to utilize succinate. Acute palmitate therefore appears to rapidly compromise or saturate complex II function and prevent utilization of substrates routed through complex II. Consistent with these in vitro findings, fresh primary microglia from 3d HFD animals failed to respond to complex II-specific mitochondrial stimuli (rotenone+succinate; *Figure 2F*) validating the findings in adult, ex vivo contexts. Consistent with the in vivo observations above, in vitro palmitate exposure decreased microglial mitochondrial length within 24 hr, indicating that fatty acid exposure itself is sufficient to trigger mitochondrial fission in a cell autonomous manner (*Figure 2G*, upper panels). This result also confirms observations by Kim et al. who observed mitochondrial fission and DRP1 phosphorylation upon 3d HFD-treated mice (*Kim et al., 2019*). Collectively, these responses were independent of any mitoSOX-detectable ROS release (*Figure 2—figure supplement 1C*), and were associated with recruitment of the mitochondrial fission regulator DRP1 (*Figure 2H*). A 24 hr exposure to the mono-unsaturated fatty acid oleate failed to elicit a comparable fission response (*Figure 2G*), and, acute palmitate exposure had no effects on microglial cytokine release in vitro (IL6, IL1β, TNFα; *Figure 2—figure supplement 1E*). Thus, acute in vitro fatty acid exposure recapitulates in vivo aMMR, including selective loss of complex II activity and lack of significant inflammatory cytokine response.

## aMMR is required for diet-induced homeostatic rewiring in vivo

We net tested whether mitochondrial dynamics are required for the in vivo metabolic responses to 3d HFD. We generated tamoxifen-inducible, microglial-*Drp1* knockout mice (*Drp1*MGKO) by crossing *Drp1*^fl/fl^ animals with a *Cx3cr1*creER transgenic line (*Goldmann et al., 2013*; *Wakabayashi et al., 2009*). *Drp1*MGKO animals were born at Mendelian ratios and grow normally. Tamoxifen injection followed by 4 weeks washout generated the intended Drp1 deletion (*Figure 3A and B*, *Figure 3—figure supplement 1A*; *Wakabayashi et al., 2009*).

Body weight, fat mass, and lean mass regulation were normal both before and after tamoxifen injection in *Drp1*MGKO mice, indicating that microglial DRP1 (and by extension microglial mitochondrial fusion) is dispensable for body weight regulation under standard conditions (*Figure 3—figure supplement 1B and C*). These data are consistent with literature (*Gao et al., 2014*). *Drp1*MGKO mice also showed normal glucose and insulin tolerance (*Figure 3C*). Importantly, whereas 3d HFD triggered post-absorptive hyperglycemia and enhanced glucose-induced insulin release in control animals, *Drp1*MGKO failed to mount a comparable whole-body metabolic response (*Figure 3D*). Thus, while microglial mitochondrial dynamics are dispensable for baseline control of body weight and glucose homeostasis, they are absolutely necessary for the very earliest metabolic anomalies induced when switching to HFD. Consistent with the data from *Figure 1*, the 3d HFD-induced hyperglycemia that is observed under overtly catabolic conditions, which we found to be microglia *independent* (*Figure 1C*), was still present in *Drp1*MGKO animals (*Figure 3D*). These data reinforce the conclusion that the metabolic responses to acute HFD are highly specific and include both microglia-dependent and independent mechanisms. Thus, microglial mitochondrial dynamics are required for the immediate in vivo homeostatic response to 3d HFD.

**Table 1.** Results of targeted metabolomics of cerebrospinal fluid (CSF).

| Metabolite | FC | log2FC | p-Value | log10(p-value) | Enrichment |
|---|---|---|---|---|---|
| Kynurenic acid | 3.56 | 1.83 | 0.24 | 0.62 | Not Sig |
| Hexadecanoic acid | 2.27 | 1.18 | 0.01 | 2.11 | HFD 3d |
| Nicotinamide-N-oxide | 0.45 | −1.14 | 0.02 | 1.71 | CT |
| Octadecanoic acid | 2.17 | 1.12 | 0.02 | 1.64 | HFD 3d |
| Serotonin | 0.51 | −0.98 | 0.95 | 0.02 | Not Sig |
| N-Methyl-4-pyridone-3-carboxamide | 1.81 | 0.85 | 0.06 | 1.21 | Not Sig |
| Methylmalonate | 1.80 | 0.85 | 0.01 | 1.86 | Not Sig |
| Tetradecanoic acid | 1.78 | 0.83 | 0.29 | 0.53 | Not Sig |
| Octanoylcamitine | 0.60 | −0.73 | 0.99 | 0.00 | Not Sig |
| Propionylcarnitine | 1.65 | 0.73 | 0.45 | 0.35 | Not Sig |
| Isovalerylcamitine | 1.65 | 0.72 | 0.02 | 1.74 | Not Sig |
| Nicotinic acid | 1.60 | 0.68 | 0.04 | 1.39 | Not Sig |
| Glutarylearnitine | 1.60 | 0.68 | 0.00 | 2.40 | Not Sig |
| Nicotinamide | 0.63 | −0.66 | 0.07 | 1.13 | Not Sig |
| 1-Methylnicotinamide | 0.66 | −0.59 | 0.31 | 0.51 | Not Sig |
| Ophthalmic acid | 0.68 | −0.56 | 0.24 | 0.62 | Not Sig |
| N-Methylserotonin | 1.41 | 0.49 | 0.17 | 0.77 | Not Sig |
| O-Acetylcarnitine | 1.38 | 0.47 | 0.62 | 0.21 | Not Sig |
| Tryptophan | 1.35 | 0.43 | 0.10 | 1.02 | Not Sig |
| 3-Hydroxyanthranillic acid | 1.34 | 0.43 | 0.20 | 0.69 | Not Sig |
| Butyrylcarnitine | 1.33 | 0.42 | 0.28 | 0.55 | Not Sig |
| Nudifioramide | 1.27 | 0.34 | 0.04 | 1.41 | Not Sig |
| Quinolinic acid | 1.21 | 0.27 | 0.22 | 0.66 | Not Sig |
| S-Adenosyl-L-homocysteine | 0.84 | −0.26 | 0.44 | 0.35 | Not Sig |
| Isobutyrylcarnitine | 1.15 | 0.21 | 0.62 | 0.20 | Not Sig |
| 2-Methylbutyrylcarnitine | 1.15 | 0.20 | 0.64 | 0.19 | Not Sig |
| Nicotinic acid mononucleotide | 0.90 | −0.15 | 0.33 | 0.49 | Not Sig |
| Hexanoylcarnitine | 1.10 | 0.14 | 0.69 | 0.16 | Not Sig |
| Nicotinamide mononucleotide | 1.10 | 0.14 | 0.95 | 0.02 | Not Sig |
| Carnitine | 1.06 | 0.08 | 0.88 | 0.06 | Not Sig |
| S-Adenosyl-L-methionine | 1.06 | 0.08 | 0.61 | 0.21 | Not Sig |
| 3-Hyroxykynurenine | 1.04 | 0.06 | 0.87 | 0.06 | Not Sig |
| Nicotinuric acid | 0.96 | −0.06 | 0.64 | 0.19 | Not Sig |
| Anthranillic acid | 1.04 | 0.05 | 0.96 | 0.02 | Not Sig |

## Microglia take up and metabolize free fatty acids

Due in part to the long isolation times required to generate pure primary adult microglia, metabolite tracing experiments on primary adult microglia are not currently feasible. We therefore chose primary murine neonatal microglia as our model of choice for more mechanistic experiments (*Valdearcos et al., 2014*). Using BODIPY fluorescence as a readout, we found that primary microglia increase lipid

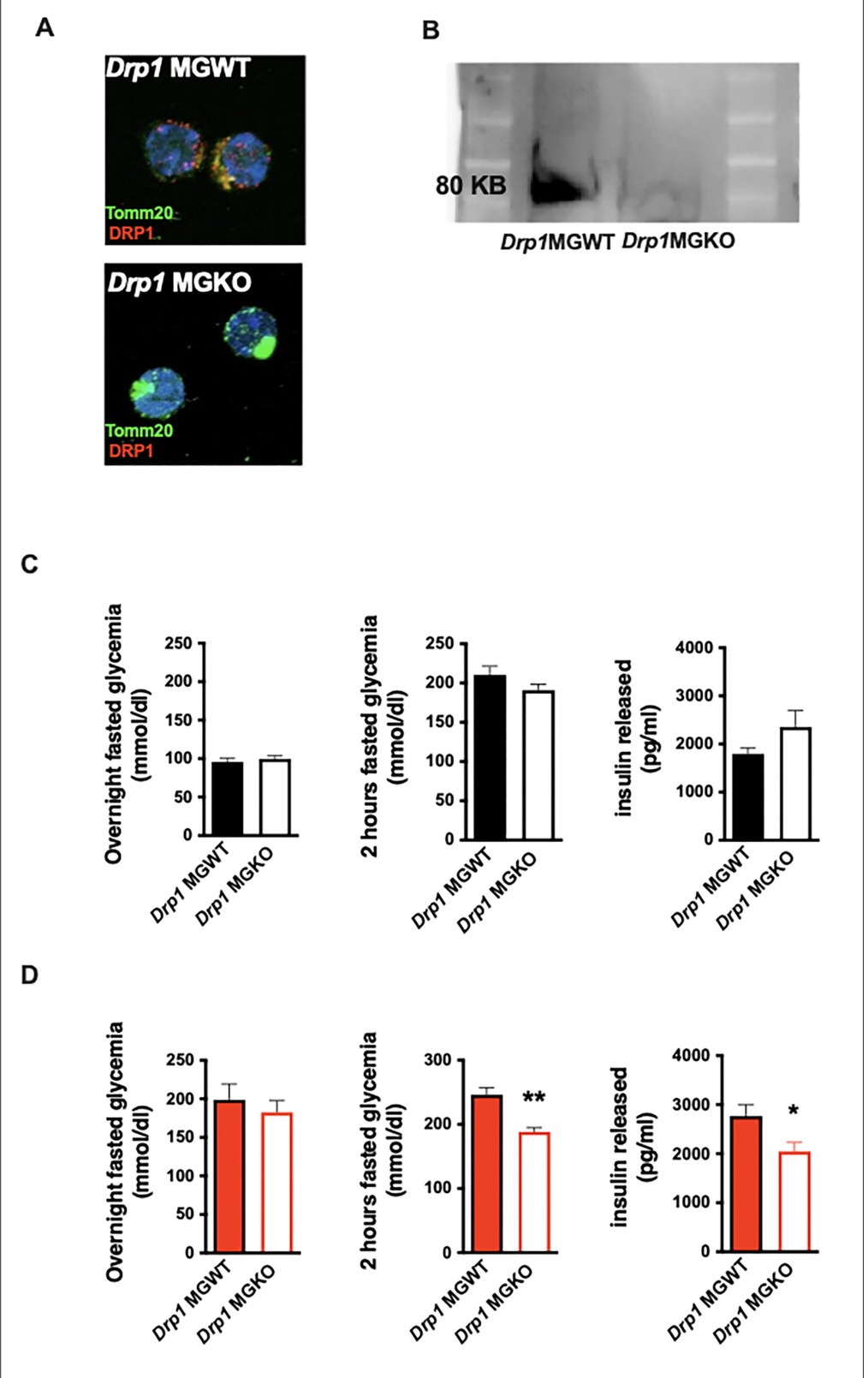

**Figure 3.** Acute microglial metabolic response (aMMR) is required for diet-induced homeostatic rewiring in vivo. (**A**) Immunostaining of TOMM20 and DRP1 on sorted microglia from mice *Drp1*MGWT or *Drp1*MGKO. (**B**) Western blot against DRP1 on sorted microglia from mice *Drp1*MGWT or *Drp1*MGKO. (**C**) Graphs showing the overnight fasted glycemia, the 2 hr fasted glycemia, and the insulin released from *Drp1*MGWT or *Drp1*MGKO fed with

*Figure 3 continued on next page*

*Figure 3 continued*

control diet (n=5–11). (**D**) Graphs showing the overnight fasted glycemia, the 2 hr fasted glycemia, and the insulin released from *Drp1*MGWT or *Drp1*MGKO fed with 3-day high-fat diet (n=5–11). Data are presented as mean ± SEM. *p<0.05, **p<0.01, ***p<0.001 as determined by two-tailed Student's t-test and two-way ANOVA followed by Bonferroni post hoc test.

The online version of this article includes the following source data and figure supplement(s) for figure 3:

**Source data 1.** Original uncropped image for *Figure 3B*.

**Source data 2.** Labelled uncropped image for *Figure 3B*.

**Figure supplement 1.** Acute microglial metabolic response (aMMR) is required for diet-induced homeostatic rewiring in vivo (supplement).

droplet numbers within 24 hr of in vitro exposure to palmitate (200 μM; *Figure 4A*), demonstrating a capacity to take up fatty acids. We therefore used stable isotopic tracer labeling and directly quantified U-[$^{13}$C]-palmitate ($^{13}$C -palmitate) processing in both control (BSA-treated) and 24 hr palmitate pre-exposed microglia. Primary microglia incubated with $^{13}$C-palmitate for 4 hr, washed and subjected to GCMS or LCMS-based metabolomics (*Figure 4B*). Focusing first on the control microglia we observed significant palmitate uptake and incorporation into downstream metabolites (*Figure 4C–F, O* and *Table 2*). This is the first definitive demonstration to our knowledge that microglia metabolize fatty acids. $^{13}$C-palmitate label was processed to palmitoylcarnitine and acetylcarnitine indicating that microglial fatty acid metabolism acts via the canonical CPT1/CPT2 pathway, moving carbons from outside the mitochondria into the inner mitochondrial matrix (*Figure 4C*, *Figure 4—figure supplement 1A*). $^{13}$C-labeling was also found in $^{13}$C-acetylserine, indicating that palmitate is processed through β-oxidation and that it contributes to the acetyl-coA pool (*Figure 4D*, *Figure 4—figure supplement 1B*). Label was detected in all measured TCA cycle intermediates (*Figure 4E*, *Figure 4—figure supplement 1C*). The labeling ratios of TCA intermediates and glutamate indicated that microglia push a substantial fraction of palmitate-derived carbons out of the TCA (alpha-ketoglutarate [a-KG] to glutamate) rather than processing them further through the whole TCA cycle (toward malate) (*Figure 4E and F*, *Figure 4—figure supplement 1D*). Thus, microglia store and metabolize palmitate toward energy and toward a unique set of anaplerotic reactions.

## Palmitate induces a novel microglial lactate/succinate/itaconate release pathway

We next evaluated the effect of palmitate *pretreatment* on the same cellular metabolic processes. Palmitate *pretreatment* of the same neonatal primary microglia model interestingly increased labeling of palmitoylcarnitine, acetylcarnitine, and acetylserine, demonstrating that 24 hr palmitate exposure accelerates β-oxidation (*Figure 4C and D*, *Figure 4—figure supplement 1A and B*). The pre-exposure enhanced m+2 label enrichment at a-KG and glutamate without impacting other TCA intermediates (*Figure 4E and F*, *Figure 4—figure supplement 1C and D*), indicating that fatty acid exposure further increases microglial substrate routing to glutamate and anaplerosis. Tracer measurements made in extracellular media conditioned from the same cells revealed that 24 hr palmitate treatment triggered release of glutamate, succinate, and intriguingly itaconate (*Figure 4G–I*), an immunomodulatory metabolite known to inhibit succinate dehydrogenase (complex II) (*Ackermann and Potter, 1949*) and recently shown to be neuroprotective (*Hooftman and O'Neill, 2019*). These findings are consistent with the complex II inhibition observed above upon 3d HFD/palmitate treatment and suggest that itaconate-dependent inhibition of complex II may be crucial in limiting the bioenergetic utilization of fatty acids by microglia, enabling their re-release and provisioning of neighboring cells. Collectively, the data demonstrate that microglia sense, process, and re-release fatty acids derived carbons in the form of usable substrates (*Figure 4O*).

We next tested whether these effects were generalizable to other carbon substrates taken up by microglia during aMMR. We repeated the experiment, replacing U-[$^{13}$C]-palmitate with U-[$^{13}$C]-glucose (*Figure 4B*). Under control conditions, microglia exhibited strong glucose uptake, glycolysis, and incorporation of glucose label into TCA intermediates (*Figure 4J*, *Figure 4—figure supplement 1E*). Substantial $^{13}$C labeling was detected in pyruvate, citrate, a-KG, malate, and again in glutamate. These data confirm that the unexpected TCA shunt toward glutamate/glutamine pathway is a unique

characteristic of the microglial metabolic fingerprint and not specific to palmitate routing. Importantly, $^{13}$C-glucose-derived metabolites were also detected *extracellularly* in the form of glutamate, itaconate, and succinate, as well as in extracellular lactate (m+3) (*Figure 4K–N*). And, palmitate pretreatment *increased* all of these signatures, including glucose uptake (*Figure 4J*), lactate labeling, and lactate release. Interestingly, as with palmitate-tracing experiments, fatty acid pretreatment increased release of tracer-labeled taconite and succinate into the extracellular media (*Figure 4L–M*). aMMR is therefore characterized by synergistic induction of a Warburg-like metabolic signature by glucose and palmitate, and significant release of carbon fuels to the cell exterior (*Figure 4N*). Collectively, these data identify microglia as a novel metabolic and neuroprotective node, able to take up harmful free fatty acids and repurpose them to fuel surrounding cells in the form of lactate and anaplerotic substrates (*Figure 4O*).

## Acute HFD induces widespread MMR and rapid modulation of spatial and learning memory

Changes in neuronal function have previously been linked to lactate provided by astrocytes (*Pellerin, 2005*). To determine if the substrates released by microglia have the potential to directly influence neurons, we incubated primary neuron cultures (*Figure 5—figure supplement 1A*) with conditioned media from the microglial $^{13}$C-glucose tracing experiments (i.e. media containing $^{13}$C-glucose-derived isotopically labeled lactate, itaconate, succinate, citrate). As a control for the direct uptake of $^{13}$C-glucose, we treated parallel neuronal cultures with the same fresh $^{13}$C-glucose tracing media originally added to the microglia. Intriguingly, and consistent with literature documenting poor direct glucose utilization by neurons (*Bouzier-Sore et al., 2006*), we found substantial m+3 lactate (as well as other metabolites) in neurons treated with microglial conditioned media, and at levels that far exceeded labeling triggered by glucose tracer alone (*Figure 5A*, middle column vs left column) (*Figure 4—figure supplement 1B*). The data indicate higher uptake of citrate and itaconate from the control microglia-conditioned media, further supporting the hypothesis that neuronal metabolism is reproducibly impacted by palmitate-triggered changes in microglial products. These data demonstrate that palmitate metabolism by microglia modulates neuronal carbon substrate use in vitro, and, they highlight the relative importance of this process compared to uptake of pure glucose. The data identify a candidate mechanism by which aMMR may alter neuronal function in vivo.

The majority of literature relating HFD-associated microglial function to metabolic regulation is focused on the hypothalamus, that exhibits an unusually leaky blood-brain barrier (*Thaler et al., 2012*; *Waterson and Horvath, 2015*). Given our findings that CSF fatty acid levels double upon acute HFD and given the dramatic metabolomic rewiring induced by microglial palmitate exposure, we asked whether aMMR might also occur in other brain regions. We FACS-sorted cortical and hippocampal microglia and tested for mitochondrial activation (MMR) in 3d HFD exposed mice. Indeed, 3d HFD exposure triggered an increase in mitochondrial membrane potential in both cortical and hippocampal microglia (*Figure 5B and C*). These data were consistent with the CSF results and suggested therefore that acute high-fat feeding might alter higher cognitive function.

Given the hippocampal signature, we tested whether 3d HFD mice showed any deleterious memory phenotype compared to chow-fed controls. Surprisingly, HFD-treated animals outperformed the controls. Using a standard Barnes Test, 3d HFD mice exhibited a faster reaction (*Figure 5D*, *Figure 5—figure supplement 2A*) suggestive of improved learning and spatial memory. Similarly, when challenged with a T-Maze, 3d HFD exposed animals exhibited heightened spatial memory (*Figure 5E*, *Figure 5—figure supplement 2E*). These phenotypes were reproducible at two different institutes (VAI, USA, and MPI-IE, Germany) and in the hands of different experimentalists (*Figure 5D and E*, *Figure 5—figure supplement 1C and D*, *Figure 5—figure supplement 2A and B*, *Figure 5—figure supplement 2E and F*). Notably, the memory differences were observed in the absence of any detectable changes to motor coordination (ROTAROD; *Figure 5F–H*, *Figure 5—figure supplement 3A*). Thus, acute high-fat feeding triggers improvements in learning and spatial memory.

To validate that these 3d HFD-induced cognitive effects were aMMR dependent, we once again repeated experiments in microglia-depleted (PLX-5622) and *Drp1*MGKO mice. Importantly, PLX-5622-treated animals failed to show any detectable memory improvements upon a 3d HFD (*Figure 5I and J*, *Figure 5—figure supplement 2C*) indicating indeed that an intact microglial compartment is necessary for acute HFD enhancement of memory function. Likewise, 3d HFD exposure failed to

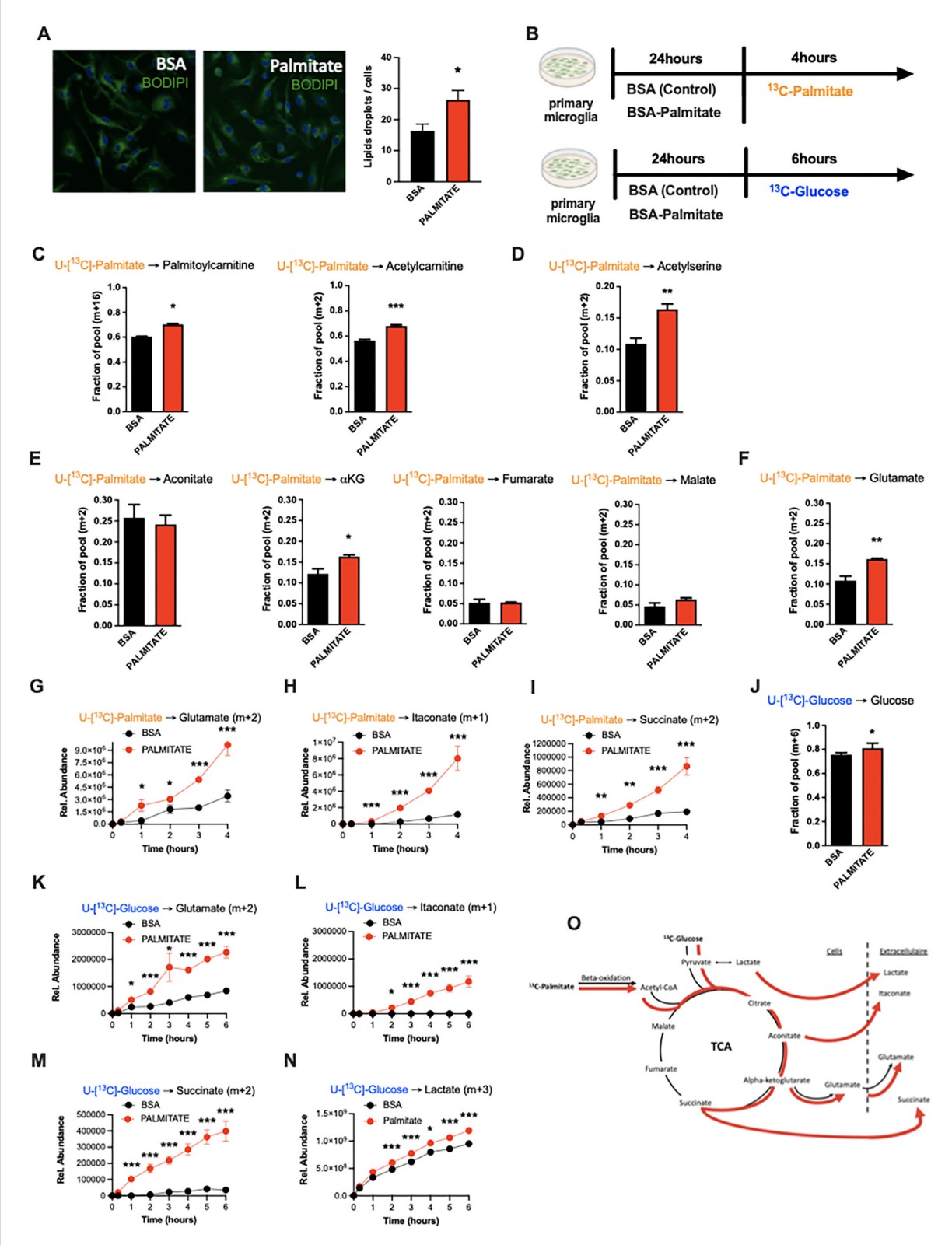

**Figure 4.** Palmitate induces a novel microglial lactate/succinate/itaconate release pathway. (**A**) BODIPY staining on primary microglia challenged for 24 hr with BSA or palmitate and the lipid droplets quantification graph (n=20). (**B**) Schematic depicting the timeline for the tracing experiments ($^{13}$C-palmitate or $^{13}$C-glucose) on primary microglial challenged for 24 hr with BSA or palmitate. (**C**) $^{13}$C-palmitate incorporation into palmitoylcarnitine (m+16) and acetylcarnitine (m+2) after 4 hr tracing experiment on primary microglia pretreated for 24 hr with BSA or palmitate (n=3). (**D**) $^{13}$C-palmitate

*Figure 4 continued on next page*

*Figure 4 continued*

incorporation into acetyl serine (m+2) after 4 hr tracing experiment on primary microglia pretreated for 24 hr with BSA or palmitate (n=3). (**E**) $^{13}$C-palmitate incorporation into aconitate, alpha-ketoglutarate, fumarate, malate (m+2) after 4 hr tracing experiment on primary microglia pretreated for 24 hr with BSA or palmitate (n=3). (**F**) $^{13}$C-palmitate incorporation into glutamate (m+2) after 4 hr tracing experiment on primary microglia pretreated for 24 hr with BSA or palmitate (n=3). (**G**) $^{13}$C-palmitate incorporation into glutamate (m+2) released during the 4 hr tracing experiment by primary microglia pretreated for 24 hr with BSA or palmitate (n=3). The results are graphed in relative abundance. (**H**) $^{13}$C-palmitate incorporation into itaconate (m+1) released during the 4 hr tracing experiment by primary microglia pretreated for 24 hr with BSA or palmitate (n=3). The results are graphed in relative abundance. (**I**) $^{13}$C-palmitate incorporation into succinate (m+2) released during the 4 hr tracing experiment by primary microglia pretreated for 24 hr with BSA or palmitate (n=3). The results are graphed in relative abundance. (**J**) $^{13}$C-glucose incorporation into the intracellular glucose pool (m+6) after 6 hr tracing experiment on primary microglia pretreated for 24 hr with BSA or palmitate (n=3). (**K**) $^{13}$C-glucose incorporation into glutamate (m+2) released during the 6 hr tracing experiment by primary microglia pretreated for 24 hr with BSA or palmitate (n=3). The results are graphed in relative abundance. (**L**) $^{13}$C-glucose incorporation into itaconate (m+1) released during the 6 hr tracing experiment by primary microglia pretreated for 24 hr with BSA or palmitate (n=3). The results are graphed in relative abundance. (**M**) $^{13}$C-glucose incorporation into succinate (m+2) released during the 6 hr tracing experiment by primary microglia pretreated for 24 hr with BSA or palmitate (n=3). The results are graphed in relative abundance. (**N**) $^{13}$C-glucose incorporation into lactate (m+3) released during the 6 hr tracing experiment by primary microglia pretreated for 24 hr with BSA or palmitate (n=3). The results are graphed in relative abundance. (**O**) Schematic depicting the metabolic pathways used by the primary microglial challenged for 24 hr with BSA (black arrow) or palmitate (red arrow) during the different tracing ($^{13}$C-palmitate or $^{13}$C-glucose). Data are presented as mean ± SEM. *p<0.05, **p<0.01, ***p<0.001 as determined by two-tailed Student's t-test and two-way ANOVA followed by Bonferroni post hoc test.

The online version of this article includes the following figure supplement(s) for figure 4:

**Figure supplement 1.** Palmitate induces a novel microglial lactate/succinate/itaconate release pathway (supplement).

trigger memory improvements in *Drp1*MGKO mice (***Figure 5K***, ***Figure 5—figure supplement 2D***) indicating that the memory enhancing response, like the glucose homeostatic effect, is microglial DRP1-dependent. Again, no mice in any treatment group showed altered motor effects by ROTAROD (***Figure 5—figure supplement 2E–J*** and ***Figure 5—figure supplement 3B and C***). Therefore, the aMMR in response to 3d HFD constitutes a generalized response that couples acute dietary changes to diverse central functions (***Figure 6***).

## Discussion

Our data identify a new role for microglial cells as a rapid metabolic sensor of dietary macronutrient composition, and as a relay that triggers cognitive and peripheral metabolic responses. One of the most intriguing results was the directionality of the rapid metabolic responses. Whereas a bulk of literature demonstrates adverse outcomes upon chronic HFD, both in the brain (microgliosis, inflammatory cytokine release, myeloid cell infiltration) and periphery (hyperglycemia, insulin resistance) (***Thaler et al., 2012***; ***Valdearcos et al., 2017***), our findings indicate that acute HFD initiates a distinct homeostatic response that supports cognitive function. Evolutionarily, there are obvious benefits to partitioning energy use toward cognition when energy needs are met. We propose that aMMR could result from direct uptake, processing, and release of fatty acid-derived carbons, and demonstrate that microglia are capable of metabolizing fatty acids toward diverse intracellular and extracellular pools. One immediate question raised by this work is the mechanistic nature of the flip from the seemingly beneficial aMMR observed here, to the well-documented inflammatory MMR triggered after 1–2 weeks of chronic HFD. One possibility is that the response requires flexibility between lipid storage and processing, and that this becomes saturated with prolonged HFD.

Of equal interest are the PLX5622 experiments and data, which demonstrate for the first time that microglial ablation has no overt effect on baseline metabolism and memory. Importantly, however, the data show that microglia are critical for optimal response to dietary change. We are aware of the fact that CSFR1 inhibiting compounds such as PLX5622 deplete not only microglia but also long living CNS macrophages in the meninges and perivascular space (***Goldmann et al., 2016***) but they are by far less numerous and placed far from neurons in the parenchyma. We demonstrate a novel role for microglia in optimizing glycemia upon short- (2 hr) but not long- (24 hr) term fasting and therefore suggest that microglia mediate a glucose-sparing function during the transition from post-prandial to fasted states. Future work mapping the downstream neuronal circuitry governing this response will be important. The overabundance of food since industrialization has skewed a significant global population toward chronic prandial and post-prandial states and away from intermittent fast-feed cycles. The finding that aMMR coordinates central metabolic balance with peripheral glucose homeostasis selectively

**Table 2.** Metabolites released in media from BSA- or palmitate (PA)-treated neurons.

| Metabolite | N | Overall, N=12* | BSA, N=6* | PA, N=6* | p-Value[†] | q-Value [‡] |
|---|---|---|---|---|---|---|
| b_DL.Lactic.Acid | 12 | 7,768,093,932.0 (566,966,988.5) | 7,315,358,105.8 (237,880,097.3) | 8,220,829,758.2 (398,345,524.3) | 0.002 | 0.064 |
| b_D..Glutamine | 12 | 92,514,878.8 (9,477,866.2) | 99,910,567.8 (6,363,882.4) | 85,119,189.7 (5,084,875.9) | 0.002 | 0.064 |
| a_Acetyl.L.carnitine | 12 | 21,534,366.5 (5,136,659.2) | 25,486,005.2 (4,313,399.0) | 17,582,727.8 (1,401,821.0) | 0.004 | 0.085 |
| b_L.Serine | 12 | 8,927,932.4 (1,999,447.4) | 10,439,581.3 (1,620,380.1) | 7,416,283.5 (827,833.2) | 0.009 | 0.13 |
| b_X4.Oxoproline | 12 | 5,158,697.7 (2,093,379.7) | 6,584,223.3 (1,496,000.9) | 3,733,172.0 (1,589,279.7) | 0.026 | 0.3 |
| b_X3.Hydroxy.2.methyl.4.pyrone.tent. | 12 | 112,324,855.5 (18,985,842.0) | 122,558,080.8 (13,194,171.3) | 102,091,630.2 (19,173,036.1) | 0.065 | 0.6 |
| a_Adipic.acid.tent. | 12 | 463,554.3 (1,036,466.7) | 0.0 (0.0) | 927,108.7 (1,359,286.8) | 0.074 | 0.6 |
| a_D..Pyroglutamic.Acid | 12 | 287,577,957.7 (32,240,216.1) | 302,927,531.8 (25,661,286.0) | 272,228,383.5 (32,600,385.7) | 0.093 | 0.6 |
| b_L.Arabinose.or.isomer. | 12 | 427,996,608.9 (18,980,682.8) | 437,653,974.3 (18,029,045.8) | 418,339,243.5 (15,611,025.3) | 0.093 | 0.6 |
| b_L.Tyrosine | 12 | 85,928,933.3 (296,055,668.3) | 345,898.5 (293,093.1) | 171,511,968.0 (418,627,435.0) | 0.13 | 0.7 |
| b_Sodium.lauryl.sulfate | 12 | 22,598.3 (76,220.7) | 0.0 (0.0) | 45,196.5 (107,496.4) | 0.2 | 0.7 |
| b_Tridecanoic.acid | 12 | 113,734.1 (302,240.1) | 0.0 (0.0) | 227,468.2 (412,217.0) | 0.2 | 0.7 |
| a_Leucine | 12 | 27,980,174.9 (7,729,502.5) | 30,862,636.3 (2,851,217.9) | 25,097,713.5 (10,167,075.1) | 0.2 | 0.7 |
| b_Glutaric.acid.tent. | 12 | 440,497,616.4 (19,061,443.4) | 448,877,445.2 (22,010,672.2) | 432,117,787.7 (12,097,256.5) | 0.2 | 0.7 |
| a_PEG.n5.tent. | 12 | 344,189.2 (776,660.6) | 4,541.3 (11,123.9) | 683,837.0 (1,024,721.6) | 0.2 | 0.7 |
| a_L.Serine | 12 | 390,966.4 (346,737.0) | 527,905.0 (409,596.4) | 254,027.8 (227,430.6) | 0.2 | 0.7 |
| b_D..Glucose.or.isomer. | 12 | 5,354,625,351.9 (372,775,074.5) | 5,219,148,234.7 (282,140,339.8) | 5,490,102,469.2 (426,687,091.4) | 0.2 | 0.7 |
| b_AICA.ribonucleotide | 12 | 64,660,392.8 (5,188,450.2) | 63,405,196.7 (3,409,871.4) | 65,915,588.8 (6,619,333.0) | 0.2 | 0.7 |
| b_neuraminic.acid.tent. | 12 | 23,066,967.0 (2,864,562.7) | 24,475,099.5 (3,412,576.8) | 21,658,834.5 (1,283,781.1) | 0.2 | 0.7 |
| a_Betaine | 12 | 3,612,697.5 (2,636,694.6) | 2,762,143.5 (2,827,681.0) | 4,463,251.5 (2,358,539.4) | 0.3 | >0.9 |
| a_DL.Arginine | 12 | 3,821,072.9 (7,222,808.1) | 2,199,664.0 (5,388,054.4) | 5,442,481.8 (8,912,410.1) | 0.3 | >0.9 |
| a_L.Threonine | 12 | 5,270,710.2 (5,478,082.2) | 6,513,108.5 (5,492,782.8) | 4,028,311.8 (5,669,696.2) | 0.4 | >0.9 |
| a_Pantothenic.acid | 12 | 22,359,057.0 (26,015,488.1) | 28,020,428.8 (25,119,049.5) | 16,697,685.2 (27,947,862.2) | 0.4 | >0.9 |
| b_L.Leucine | 12 | 6,646,608.2 (3,705,017.3) | 7,399,027.0 (3,879,174.6) | 5,894,189.3 (3,713,895.4) | 0.4 | >0.9 |
| b_Crotonic.acid | 12 | 16,108,825.9 (23,803,380.1) | 23,947,637.5 (26,242,912.3) | 8,270,014.3 (20,257,315.3) | 0.4 | >0.9 |
| b_Succinic.acid | 12 | 407,804.1 (869,276.7) | 109,080.2 (267,190.7) | 706,528.0 (1,173,394.8) | 0.5 | >0.9 |
| a_L.Lysine | 12 | 24,416,825.1 (28,518,776.0) | 34,566,558.7 (36,434,829.3) | 14,267,091.5 (14,648,122.2) | 0.5 | >0.9 |
| b_Pyridoxal.tent. | 12 | 777,482.7 (1,211,214.3) | 598,171.7 (398,509.7) | 956,793.7 (1,729,598.9) | 0.5 | >0.9 |
| b_X2.C.Methyl.D.erythritol4.phosphate.tent. | 12 | 643,051,857.5 (148,219,463.1) | 606,454,245.0 (172,899,107.7) | 679,649,470.0 (123,382,309.5) | 0.5 | >0.9 |
| b_X2.Methylsuccinic.acid.tent. | 12 | 287,186,644.6 (143,335,650.3) | 264,074,195.0 (155,198,662.0) | 310,299,094.2 (140,821,055.1) | 0.5 | >0.9 |
| b_Pyruvic.acid.tent. | 12 | 150,010,045.5 (9,715,420.7) | 147,364,165.0 (6,350,912.8) | 152,655,926.0 (12,268,697.7) | 0.5 | >0.9 |
| a_Niacinamide | 12 | 1,384,815.1 (2,023,883.3) | 1,614,438.5 (2,373,808.4) | 1,155,191.7 (1,802,752.7) | 0.5 | >0.9 |
| a_X6.Methoxyquinoline.tent. | 12 | 7,956,394.3 (26,119,894.0) | 15,246,370.2 (37,028,395.4) | 666,418.3 (1,516,716.5) | 0.5 | >0.9 |
| a_L..Methionine | 12 | 788,318.3 (1,316,035.4) | 642,601.0 (747,341.7) | 934,035.5 (1,789,080.3) | 0.5 | >0.9 |
| b_X4.Hydroxyquinoline | 12 | 2,294,626.8 (5,198,100.9) | 3,743,521.2 (7,079,171.4) | 845,732.5 (2,071,613.1) | 0.6 | >0.9 |
| a_X2.2.6.6.Tetramethyl.4.piperidinol.tent. | 12 | 3,349,971.7 (5,949,414.1) | 4,474,480.0 (7,959,519.7) | 2,225,463.3 (3,388,416.0) | 0.6 | >0.9 |
| a_N.N.Diethylethanolamine.tent. | 12 | 3,251,751.6 (3,370,326.9) | 2,444,947.7 (2,152,767.5) | 4,058,555.5 (4,335,133.4) | 0.6 | >0.9 |
| b_L.Isoleucine | 12 | 7,328,303.6 (7,854,185.2) | 8,665,141.0 (7,496,666.0) | 5,991,466.2 (8,673,233.1) | 0.7 | >0.9 |
| a_X6.Methylquinoline.tent. | 12 | 385.0 (699.5) | 274.2 (671.6) | 495.8 (772.0) | 0.8 | >0.9 |
| a_Hypoxanthine | 12 | 38,073.7 (83,225.3) | 44,836.3 (109,826.1) | 31,311.0 (55,377.5) | 0.8 | >0.9 |
| b_p.Toluenesulfonic.acid.tent. | 12 | 3,372,499.5 (6,572,551.8) | 3,393,943.8 (5,259,090.6) | 3,351,055.2 (8,208,375.3) | 0.8 | >0.9 |

*Table 2 continued on next page*

*Table 2 continued*

| Metabolite | N | Overall, N=12* | BSA, N=6* | PA, N=6* | p-Value[†] | q-Value[‡] |
|---|---|---|---|---|---|---|
| b_Urocanic.acid.tent. | 12 | 1,399.0 (2,534.9) | 1,818.7 (2,821.3) | 979.3 (2,398.9) | 0.8 | >0.9 |
| a_Pyridoxal.tent. | 12 | 1,928,431.0 (3,031,562.9) | 2,203,965.7 (3,922,077.7) | 1,652,896.3 (2,157,295.3) | 0.8 | >0.9 |
| b_X3.Methyl.2.oxovaleric.acid | 12 | 1,101,172.5 (3,039,252.9) | 282,158.3 (278,456.3) | 1,920,186.7 (4,316,724.3) | 0.8 | >0.9 |
| b_L.Methionine | 12 | 747,354.3 (822,900.9) | 676,740.7 (845,132.3) | 817,968.0 (873,813.0) | 0.8 | >0.9 |
| b_L.Tryptophan | 12 | 198,603.2 (296,935.2) | 272,248.2 (407,584.4) | 124,958.2 (121,792.6) | 0.8 | >0.9 |
| a_Indole.3.acrylic.acid | 12 | 2,553,283.9 (3,140,722.0) | 2,083,026.7 (2,110,564.8) | 3,023,541.2 (4,088,504.8) | 0.8 | >0.9 |
| b_X4.Dodecylbenzenesulfonic.acid.tent. | 12 | 10,280,591.5 (7,701,423.0) | 8,121,727.2 (4,928,645.2) | 12,439,455.8 (9,747,256.3) | 0.8 | >0.9 |
| a_N.Acetylputrescine | 12 | 386,960.3 (468,563.5) | 391,056.2 (550,659.6) | 382,864.3 (423,966.5) | 0.9 | >0.9 |
| a_D..Proline | 12 | 661,884.2 (1,149,335.2) | 957,007.0 (1,564,926.8) | 366,761.3 (498,103.3) | 0.9 | >0.9 |
| a_X4.Aminonicotinic.acid.or.isomer. | 12 | 126,303.5 (225,453.7) | 94,034.3 (196,574.4) | 158,572.7 (265,864.7) | >0.9 | >0.9 |
| a_Choline | 12 | 174,331,021.6 (21,492,889.5) | 171,722,892.2 (20,170,821.4) | 176,939,151.0 (24,353,435.2) | >0.9 | >0.9 |
| b_Acetoacetic.acid | 12 | 319,609,159.2 (14,397,923.5) | 321,712,028.7 (13,338,036.2) | 317,506,289.7 (16,356,781.0) | >0.9 | >0.9 |
| a_Isoleucine | 12 | 30,943,808.3 (15,505,575.1) | 29,880,732.5 (15,885,905.7) | 32,006,884.2 (16,548,594.4) | >0.9 | >0.9 |
| b_L.Phenylalanine | 12 | 9,511,810.9 (5,608,112.4) | 10,045,803.3 (6,593,135.4) | 8,977,818.5 (5,003,823.0) | >0.9 | >0.9 |
| a_Creatine | 12 | 137,229.8 (326,719.4) | 112,430.3 (275,396.9) | 162,029.2 (396,888.8) | >0.9 | >0.9 |
| b_D..Fructose.or.isomer. | 12 | 1,376,324,172.0 (124,013,739.0) | 1,382,734,756.2 (112,787,783.2) | 1,369,913,587.8 (144,965,454.4) | >0.9 | >0.9 |
| b_Folic.acid | 12 | 3,351,599.5 (3,207,090.2) | 4,052,769.5 (4,167,729.0) | 2,650,429.5 (2,019,416.2) | >0.9 | >0.9 |
| b_Propylparaben.or.isomer. | 12 | 255,003.3 (618,051.8) | 190,425.0 (466,444.1) | 319,581.7 (782,812.0) | >0.9 | >0.9 |

*Mean (SD).

[†]Wilcoxon rank sum test; Wilcoxon rank sum exact test.

[‡]False discovery rate correction for multiple testing.

during the prandial/post-prandial phase adds nuance to a significant literature distinguishing these physiological contexts (*Dakic et al., 2017*; *Havel, 2001*; *Zeltser et al., 2012*). From an evolutionary perspective it is easy to rationalize why initial periods of plenty would be mechanistically coupled to maximizing central functions.

One question raised by these findings is if there is a hierarchy for substrate sharing between glial cells (microglia, tanycytes, oligodendrocytes, and astrocytes) and neurons. Our data indicate that microglia provide several TCA-associated substrates into the extracellular milieu, and, that palmitate/HFD increases this activity. Tanycytes and astrocytes have both been documented to release select metabolites into the extracellular environment (*García-Cáceres et al., 2019*; *Barca-Mayo and López, 2021*). The experiments presented here do not completely rule out a contribution of these or cell types in coupling acute HFD intake to memory and learning. Given that CSF production is only loosely compartmentalized (e.g. relative to the highly physically structured control of blood), we suspect a cooperative model over hierarchical, serial, or synergistic metabolic compartmentalization models. Other new questions relate to the depth or breadth of other cognitive processes that might also be influenced by acute HFD. Here, we demonstrate that aMMR involves release of at least four important anaplerotic substrates, and that aMMR is detectable from the hypothalamus to the hippocampus and cortex. Increased lactate has been shown to enhance fear memory in a glutamate-associated mechanism (*Ikeda et al., 2021*). Both lactate and glutamate are increased upon 3d HFD suggesting that aMMR may comprise a much more complex behavioral response.

In addition to identifying novel substrate routing patterns in microglia, our data demonstrate that mitochondrial fission is necessary for aMMR. Fatty acid-triggered fission has been observed in several cell types (*Buck et al., 2016*) including neurons and glial cells (*Schneeberger et al., 2013*; *Ramírez et al., 2017*; *Kim et al., 2019*). Work by *Kim et al., 2019*, suggests that the aMMR characterized here may require UCP2; those authors demonstrated UCP2-dependent mitochondrial changes in short- and long-term HFD exposed hypothalamic microglia. Collectively, the two studies argue

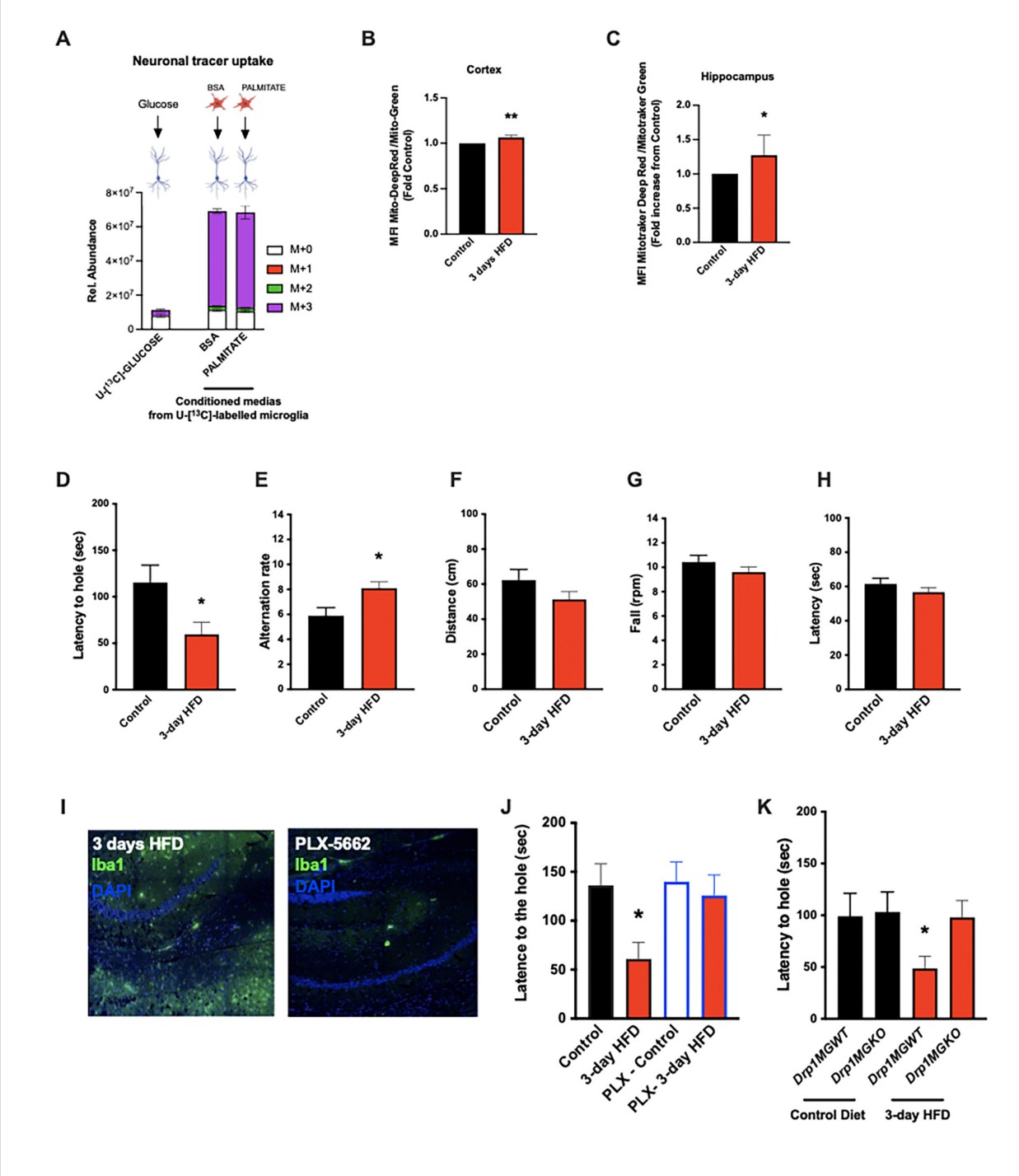

**Figure 5.** Acute high-fat diet (HFD) induces widespread microglial metabolic response (MMR) and rapid modulation of spatial and learning memory. (**A**) Primary microglial cell media was collected after the $^{13}$C-glucose tracing (containing $^{13}$C-lactate released by microglia challenged with BSA or palmitate) and incubated for 4 hr with primary neurons, the graph shows the $^{13}$C-lactate incorporation in the neurons in relative abundance, as control, primary neurons were incubated directly with $^{13}$C-glucose (n=6). (**B**) Graph showing the ratio MitoTracker Deep Red/ MitoTracker Green from sorted cortical microglial cells of C57Bl6/J male fed with a control diet (Control) or fed with high-fat diet for 3 days (3-day HFD) (n=5–12). (**C**) Graph showing the ratio MitoTracker Deep Red/ MitoTracker Green from sorted hippocampal microglial cells of C57Bl6/J male fed with a control diet (Control) or fed with 3-day HFD (n=5–12). (**D**) Graph showing the latency during the Barnes Test from mice fed with normal diet (Control) or 3-day HFD (n=11). The test was performed in the VAI animals facility (USA). (**E**) Graph showing the alternation during the T Maze Test from mice fed with normal diet (Control) or 3-day HFD (n=11). The test was performed in the VAI animals facility (USA). (**F**) Graph showing the distance walked during the ROTAROD test from mice fed with normal diet (Control) or 3-day HFD (n=11). (**G**) Graph showing the number of turn before the mice fall during the ROTAROD test from mice fed with

*Figure 5 continued on next page*

*Figure 5 continued*

normal diet (Control) or 3-day HFD (n=11). (**H**) Graph showing the latency during the ROTAROD test from mice fed with normal diet (Control) or 3-day HFD (n=11). (**I**) Microglial staining with Iba1 (green) in the hippocampus slices from mice fed with 3 days HFD or mice depleted from their microglial cells with 1 week control diet complexed with PLX-5662 prior the 3 days HFD (PLX-5662) (n=5). (**J**) Graph showing the latency during the Barnes Test from mice fed with normal diet (Control) (n=6), mice fed with 3-day HFD (n=6), mice depleted from their microglial cells with 1 week control diet complexed with PLX-5662 (PLX-Control) (n=8) or mice depleted from their microglial cells with 1 week control diet complexed with PLX-5662 prior the 3 days HFD (PLX-3-day HFD) (n=8). (**K**) Graph showing the latency during the Barnes Test from *Drp1*MGWT or *Drp1*MGKO mice fed with normal diet (Control diet) or with 3-day HFD (n=11). Data are presented as mean ± SEM. *p<0.05, **p<0.01, ***p<0.001 as determined by two-tailed Student's t-test and two-way ANOVA followed by Bonferroni post hoc test.

The online version of this article includes the following figure supplement(s) for figure 5:

**Figure supplement 1.** Acute high-fat diet (HFD) induces widespread microglial metabolic response (MMR) and rapid modulation of spatial and learning memory (supplement).

**Figure supplement 2.** Acute HFD induces widespread microglial metabolic response (MMR) and rapid modulation of spatialand learning memory (supplement — individual replicate data).

**Figure supplement 3.** Acute high-fat diet (HFD) doesn't affect the motor coordination.

against astrocytes being the only cells in their ability to use fatty acids and release fatty acid-derived substrates for use by neurons and neighboring cells (*Westergaard et al., 1995*). Studies have identified stearate and palmitate in the CSF of patients with chronic obesity and with diabetes, reports that highlight the importance of these findings (*Melo et al., 2020*). While a systematic dissection of the roles of HFD-regulated CSF metabolites (direct [diet containing] and indirect [secondary]) is beyond the scope of this study, though they represent priority areas for future investigation. This is particularly

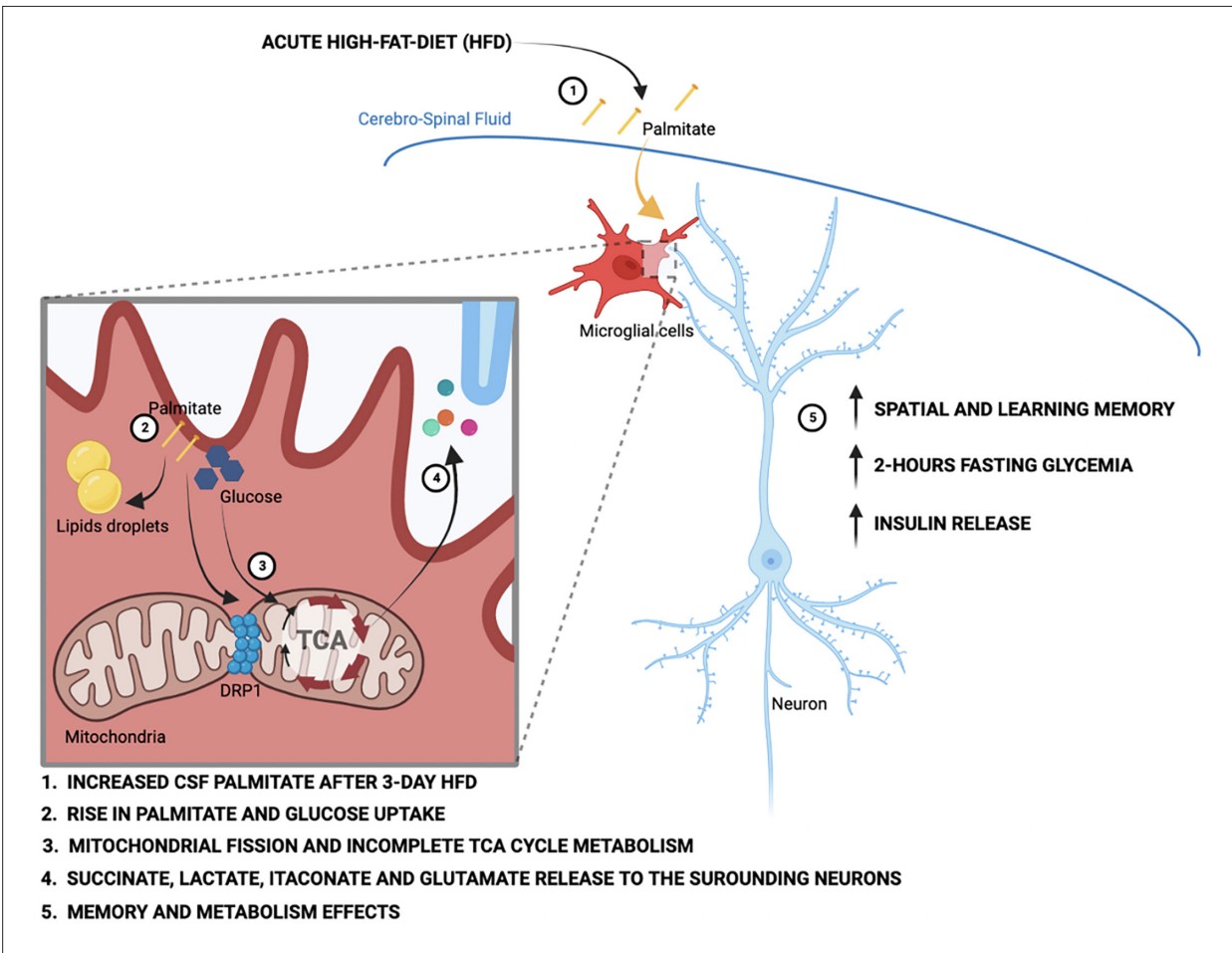

**Figure 6.** Schematic depicting the acute high-fat diet impact on microglial cells via the metabolic pathways wiring.

true given the wide range of fatty acid-specific biological effects in the literature, and the potential for combinatorial interactions.

Finally, this study demonstrates unappreciated plasticity and complexity in microglial cellular metabolism. Rather than being glucose-constrained, we reveal that primary microglia from across the brain readily take up, metabolize, and store fatty acids and that fatty acids are metabolized toward bioenergetic and acetylation reactions (acetyl-coA). They prove for the first time that substantial β-oxidation occurs in microglial cells even in the presence of high extracellular glucose (*Kalsbeek et al., 2016*). The coordinate observations of intra- and extracellular succinate, itaconate accumulation, and relative complex II inhibition indicate that palmitate metabolism drives feedforward inhibition of complex II via itaconate production and shunting of TCA intermediates out of the cycle toward glutamate. Further examination of the role of glutamine as a fuel and anapleurotic regulator of cellular function is therefore warranted. At least under the standard conditions used here, microglia appear metabolically wired to also release lactate, succinate, glutamate, citrate, and itaconate. The idea is consistent with the observed lack of substantial intracellular lipid accumulation, respiration, or ROS buildup upon prolonged palmitate exposure. Such substrate release could mediate the learning and memory effects that accompany aMMR; they are consistent with the data of other studies that have examined metabolite associations with learning and memory (*Morgunov et al., 2020*; *Xiong et al., 2023*; *Serra et al., 2022*; *Cline et al., 2012*).

## Materials and methods
### Experimental animals
All animal studies and experimental procedures were approved by the Van Andel Institute Institutional Animal Care and Use Committee under protocol #22-07-027. All mice were housed in a temperature-controlled environment (23°C) with a 12 hr light and 12 hr dark (07.00–19.00) photoperiod. Animals were provided standard chow diet (Research Diets #5021; 23.6% calories from fat; Mouse Breeder Diet #6539) or HFD (60% fat; Research Diets, Rodent Chow #D12492) and water ad libitum unless otherwise stated.

For the experiments performed on non-modified mice, C57BL/6J were purchased from Jackson Laboratory. *Drp1* floxed mice (*Drp1^fl/fl^*) and Cx3cr1^CreER^ mice (B6.129P2(C)-Cx3cr1^tm2.1(cre/ERT2)Jung^/J) were purchased from Jackson Laboratory. Mice expressing tamoxifen-inducible Cre recombinase (CreERT2) in cells expressing CX3CR1 (Cx3cr1-cre:ERT2) were crossed with mice harboring conditional alleles Drp1 (*Drp1^fl/fl^*). For induction of Cre recombinase, 5- to 6-week-old *Drp1^MGKO^* (*Drp1^fl/fl^*-Cx3cr1-cre) mice and their littermate control were treated with 4 mg tamoxifen (TAM, Sigma) solved in 200 μl corn oil (Sigma) injected subcutaneously at two time points 48 hr apart. HFD was started 4 weeks from the last injection to allow the replacement of peripheral monocytes (*Goldmann et al., 2013*). Mice were sacrificed at different time points during the diet. Only males were used in this study.

### Microglial depletion
Microglia were depleted in all cases by feeding mice custom diet (Research Diets) containing the CSF1R inhibitor PLX5622 (Plexxikon) formulated in the exact diets indicated above. The compound-containing diets have the new order numbers (D12450: PLX5622-containing standard diet and D21021905: PLX5622-containing HFD). PLX5622 was included to achieve a dose of 1.2 g/kg based on average adult C57BL/6J food intake. Animals were administered PLX5622-containing diet for 7–9 days without observable impact on the body weight or food intake (*Figure 1—figure supplement 2A and B*), using protocols adopted from *Feng et al., 2017*; *George et al., 2019*.

### Microglial cultures
Eight to 10 mice (1–3 days of age, mixed sex) from C57BL/6J mice were used for primary microglial culture. In brief, brains were dissected and placed in a small culture dish that contained a small volume of Dulbecco's modified eagle medium (DMEM). The meningeal lining (dura and arachnoid layers) were then gently removed using two small straight forceps under a magnifying glass. The cleaned brain was then placed in a new culture dish minced into a fine slurry using a pair of corneal scissors. Suspended cells were filtered (70 μm) and centrifuged for 5 min at 1000 rpm and the pellet was resuspended and plated on poly-D-lysine (Cat# P6407, Sigma)-coated 75 cm² flasks with DMEM containing 1%

penicillin-streptomycin, 10% fetal bovine serum. Seven to 10 days later, the flasks were stimulated for 3 days with 30 ng/ml M-CSF (Cat# 315-02, Peprotech) and then, shaken (200 rpm) for 3 hr (37°C) to specifically release microglia.

For conditioned media experiments, microglial cells were incubated with DMEM (Gibco) without lactate completed with BSA-conjugated palmitate or control BSA. Conditioned media was collected after the incubation, centrifuged 15 min at 300×$g$ (4°C), and the supernatant transferred and frozen in a fresh tube avoiding the cells and debris pellet. Samples were immediately snap-frozen or used for the neurons incubation.

## Embryonic cortical neuron culture protocol

Primary neuron cultures were prepared from embryonic day 18 CD1 mice. Brains were gently removed from the embryos and placed into a Petri dish filled with ice-cold, sterile Hibernate Medium (Cat# A1247601, Gibco). Hemispheres were gently separated, and the meninges, thalamus, striatum, brainstem, and hippocampus were removed. Cortical tissue was isolated and cut into 1 mm$^3$ segments. The cortical tissue from each brain was pooled and digested in papain solution (20 U/ml Cat# LS003126, Worthington) and then treated with DNase I (Cat# LS006563, Worthington) to remove residual DNA. The tissue was then washed with pre-warmed Neurobasal media (Cat# 21103049, Gibco), mechanically dissociated, and strained through a 40 µm cell strainer. The cell suspension was pelleted at 1000 rpm for 5 min, resuspended in 2 ml of neuron media (Neurobasal media containing 1% B27, 2 mM GlutaMAX, and penicillin-streptomycin) and gently mixed. The dissociated neurons were seeded on poly-D-lysine-coated six-well culture plates at 1 million cells/well. On DIV3, cytosine arabinoside (1-β-D-arabinofuranosylcytosine) was added to a final concentration of 5 µM to curb glial proliferation. The neurons were maintained until DIV21 by replacing 1/3 volume of media with fresh neuron media every 5 days.

## Microglia isolation

Mice were anesthetized and blood was collected by ventricular puncture. Mice were then perfused with phosphate-buffered saline (PBS). For microglia isolation, the hypothalamus, cerebellum, hippocampus, and cortex were dissected from the brain and manually dissociated in HBSS buffer. A cell suspension was prepared with a continuous 37% Percoll gradient at 2300 rpm for 30 min without brake, then cells were washed with 1× PBS and resuspended in FACS buffer and non-specific binding to Fc receptors was blocked by incubation with anti-CD16/32 antibody (BD Pharmingen, Cat# 553141) and the FACS antibodies BV711-conjugated anti-CD11b and Pacific blue-conjugated anti-CD45 antibodies (Cat# 101241, Cat# 103126, BioLegend).

## scRNAseq

For the scRNAseq experiments, pure hypothalamic microglial cells were sorted and processed as previously described (*Tay et al., 2017*; *Tay et al., 2018*).

Briefly, brains were isolated and gently homogenized and resuspended in 20 ml of ice-cold extraction buffer (1× HBSS, 1% fetal calf serum, 1 mM EDTA). Microglial cells were extracted in 5 ml of 37% isotonic Percoll. After staining, single CD45$^{lo}$, CD11b$^+$ microglial cells were FACS-sorted directly in 192-well plates, containing 192 unique barcodes, for scRNAseq. The CEL-Seq2 method was used for single-cell sequencing (*Hashimshony et al., 2016*; *Herman and Grün, 2018*). Thirteen libraries (2492 single cells) were sequenced on an Illumina HiSeq 2500 system, as pair-end multiplexing run, with 50–75 bp read length.

## scRNAseq analysis

The scRNAseq libraries were analyzed using the 'scRNAseq' module from snakePipes v.1 (*Bhardwaj et al., 2019*). Within the snakePipes pipeline, trimming of barcodes was achieved using Cutadapt v.1.9.1 (*Martin, 2011*). Quality of the reads was evaluated with FastQC (https://www.bioinformatics.babraham.ac.uk/projects/fastqc/). Mapping to the GRCm38/mm10 mouse genome was performed using the STAR aligner v.2.5.3a (*Dobin et al., 2013*), and read counts on features were quantified by featureCounts v.1.6.1 (*Liao et al., 2014*). Mapped and counted read were next analyzed with RaceID3 v.0.2.6, with default parameters (*Herman and Grün, 2018*; *Grün, 2020*). Batch correction by plate and preprocessing of data were completed within the RaceID3 pipeline. We included in our analysis,

all microglial cells from control (CT) and 3-day high-fat diet-treated (3d_HFD) mice. We filtered out all the cells with a total transcript count per cell >1000. Dimensional reduction of transcriptomic profiles was then preformed with Uniform Manifold Approximation and Projection. As a comparative analysis, we also added to our dataset previously published single microglial cells from EAE, and matched controls mice (CT_EAE) (*Melo et al., 2020*).

## qPCR

For the qPCR experiments, pure microglial cells were sorted, and the RNA was extracted with TRIzol (ImGen protocol).

|  | Foward | Reverse |
|---|---|---|
| Tmem119 | GTGTCTAACAGGCCCCAGAA | AGCCACGTGGTATCAAGGAG |
| P2ry12 | CAAGGGGTGGCATCTACCTG | AGCCTTGAGTGTTTCTGTAGGG |
| Arg1 | GGAAATCGTGGAAATGAG | CAGATATGCAGGGAGTCACC |
| S100B | AACAACGAGCTCTCTCACTTCC | CTCCATCACTTTGTCCACCA |
| RPL19 | GAAGGTCAAAGGGAATGTGTTCA | CCTTGTCTGCCTTCAGCTTGT |

## Mitochondrial function by cytometry

For mitochondrial activity experiments, cell suspension was incubated with MitoTracker Green FM (Invitrogen, Cat# M7514) and MitoTracker DeepRed FM (Invitrogen, Cat# 22426) for 30 min at 37°C before FACS acquisition. For the electron transport chain experiments (ETC), the experiment was based on the *Salabei et al., 2014*. Cell suspensions were incubated with the mitochondrial probe tetramethylrhodamine TMRM 10 mM (Abcam, Cat# ab228569) for 30 min at 37°C before FACS acquisition. For challenging the ETC, the cell pellet was resuspended in 500 µl of warm MAS buffer solution (*Salabei et al., 2014*) + 1 nM Plasma Membrane Permeabilizer (Agilent Seahorse XF PMP) to permeabilize the cells. Microglial cells were gated from CD45low-CD11b+ cells followed by singlet after forward and side scatter pattern. They were incubated each 90 s by the following drugs: 0.5 µl of 100 µM rotenone (Sigma), 2 µl of 2.5 M succinate pH 7.4 (succinic acid Sigma) and 0.5 µl of 1 mM antimycin (Sigma). Cytometry was performed on Fortessa (BD Bioscience) and analyzed with FlowJo v.10 (Treestar). Microglial cells were gated from CD45low-CD11b+ cells followed by singlet after forward and side scatter pattern.

## Confocal microscopy analysis

Mice were anesthetized and transcardially perfused with 0.9% saline followed by fixative (4% paraformaldehyde [PFA]). Brains were incubated 24 hr with the fixative, then 48 hr with 20% sucrose and finally fixed with Polyfreeze (Sigma). Frozen brains were sliced into 7 µm and incubated in 0.1%PBST+5% BSA with anti-Iba1 antibody (diluted 1:300, Abcam; Cat# ab178846) and/or anti-GFP antibody (diluted 1:200, Cell Signaling; Cat# 2255), or anti-DRP1 antibody (diluted 1:50 Cell Signaling; Cat# 8570). After several washes with PBS, sections were incubated in the appropriated secondary antibodies (Thermo Fisher) for 1 hr at room temperature, then rinsed in PBS three times 10 min each time, and flat-embedded in mounting media with DAPI (Fluoroshield, Abcam; Cat#104139).

For the colocalization experiment, microglia were isolated from 10- to 12-week-old *Drp1*KO mice and their littermate controls, immediately fixed in PFA and stained with DRP1 (diluted 1:50 Cell Signaling). The Pictures acquisition were performed with the Confocal LSM780 (Zeiss).

## Glucose and insulin tolerance

OGTTs were performed in 16 hr fasted animals. After determination of basal blood glucose levels, each animal received a gavage of 1 g/kg glucose (Sigma) and blood glucose levels were measured at –30, 0, 15, 30, 45, 60, 90, and 120 min after glucose administration using a glucometer (Accu-Check, Roche). In the same cohort of mice, blood samples were also collected for determination of circulating insulin levels during the OGTT at –30, 15, 30, and 60 with the ELISA insulin assay kit (Mercodia).

ITTs were performed in 2 hr fasted animal. After determination of basal blood glucose levels, each animal received an i.p. injection of insulin, 0.75 U/kg (Actrapid, Novo Nordisk). Blood glucose levels were measured at –30, 0, 15, 30, 45, 60, 90, and 120 min after insulin administration.

## Body composition

Body composition was measured in vivo by MRI (EchoMRI; Echo Medical Systems, Houston, TX, USA).

## Behavioral tests

Mice were fasted at 7:00 AM and the behavioral tests were performed between 8:00 AM and 2:00 PM. ROTAROD, Barnes, and T Maze behavioral tests were performed as previously described (*Shiotsuki et al., 2010*; *Illouz et al., 2020*).

## Microglial analysis

To analyze microglial number and size in the ARC, VMH, and cortex, TdTomato positive cells were counted manually from four hypothalamic level-matched sections per animal using ImageJ software and microglial cell size was measured using a thresholding parameter on ImageJ software. A total of 40 cells per section were used to determine the size in all regions.

## Measures of mitochondrial length

Primary microglial cells were seeded overnight in poly-D-lysine-coated cell culture chambered coverslips (Lab-Tek Cat# 155411, Thermo Scientific) at $1 \times 10^4$ cells/well density in 100 µl microglial growth media. Cells were incubated at 37°C for 2, 6, or 24 hr in microglial growth media including 100 µM BSA or 100 µM palmitate complexed with BSA (*Valdearcos et al., 2014*) or 100 µM oleate complexed with BSA or 1 µg/ml LPS (*Katoh et al., 2017*). One half-hour before the end of the incubation, 200 nM MitoTracker Green (Invitrogen, Cat# M7514) was added in the media. Cells were washed three times in incubation media and the mitochondrial network was observed on a Confocal LSM 780 (Zeiss). After image acquisition, 5 µM MitoSox (Invitrogen, Cat# M36008) was added to the incubation media for 10 min. Cells were washed three times with media and fixed with 4%PFA at 37°C for 15 min. MitoSox staining was observed on a Confocal LSM 780 (Zeiss). Alternatively, cells were permeabilized with 0.2% Triton in PBS, incubated with blocking buffer (PBS+5% FCS+0.1% Tween) for 1 hr and stained with anti-Iba1 antibody (diluted 1:300, Abcam; Cat# ab178846) and/or anti-TOM20 antibody (diluted 1:1000, Santa Cruz; Cat# sc177615) and/or anti-DRP1 antibody (diluted 1:50 Cell Signaling; Cat# 8570). After three washes with PBS, sections were incubated with appropriate secondary antibodies (Thermo Fisher) for 1 hr at room temperature, rinsed three times 10 min in PBS, and image acquisition performed on a Confocal LSM780 (Zeiss).

## Cytokine measurement

Media from primary microglia incubated with BSA or palmitate was collected and processed using the mouse Custom Panel Standard kit (#93976 LegendPlex).

## Seahorse measurement

Primary microglia were incubated with BSA or palmitate conjugated to BSA (Agilent Seahorse Palmitate-BSA conjugation protocol) for 24 hr then incubated in the Seahorse XF (±0.25 mM succinate) and processed using the Cell Mito Stress Test Kit (#103015 Agilent).

## Stable isotope labeling and metabolomics

Metabolomics was performed on microglia by first washing microglia primary cell cultures with PBS and re-cultured in DMEM (lacking glucose) containing 10% dialyzed FBS and uniformly labeled [$^{13}$C]-glucose (Cambridge Isotope Laboratories). Microglia ($2 \times 10^6$ per well in six-well plates) were cultured in $^{13}$C-containing medium for up to 6 hr. For cellular media samples, 10 µl of media were taken at indicated time points and centrifuged to remove cells, with 10 µl of media used for metabolite analysis. For uniformly labeled [$^{13}$C]-palmitate tracing analysis, sodium [$^{13}$C]-palmitate was conjugated to BSA following Agilent Seahorse Palmitate-BSA conjugation protocol. For [$^{13}$C]-palmitate tracing, microglial primary cell cultures were first cultured for 24 hr with BSA control or $^{12}$C-palmitate (200 µM) in DMEM containing 10% dialyzed FBS. Following 24 hr incubation, the media was removed, and microglial cells washed with PBS followed by culture in DMEM with 10% dialyzed FBS and blank mM [$^{13}$C]-palmitate for 4 hr and cell and media samples collected as before.

For gas chromatography coupled to mass spectrometry (GC-MS) metabolites were extracted using ice-cold 80% methanol, sonicated, and then D-myristic acid was added (750 ng/sample) as an internal

standard. Dried samples were dissolved in 30 µl methozyamine hydrochloride (10 mg/ml) in pyridine and derivatized as tert-butyldimethylsilyl (TBDMS) esters using 70 µl *N*-(*tert*-butyldimethylsilyl)-*N*-methyltrifluoroacetamide (*Faubert et al., 2014*). For metabolite analysis, an Agilent 5975C GC-MS equipped with a DB-5MS+DG (30 m × 250 µm × 0.25 µm) capillary column (Agilent J&W, Santa Clara, CA, USA) was used. All data were collected by electron impact set at 70 eV. A total of 1 µl of the derivatized sample was injected in the GC in splitless mode with inlet temperature set to 280°C, using helium as a carrier gas with a flow rate of 1.5512 ml/min (rate at which myristic acid elutes at 17.94 min). The quadrupole was set at 150°C and the GC/MS interface at 285°C. The oven program for all metabolite analyses started at 60°C held for 1 min, then increased at a rate of 10 °C/min until 320°C. Bake-out was at 320°C for 10 min. Sample data were acquired both in scan (1–600 m/z) modes. MassHunter software (v.10, Agilent Technologies) was used for peak picking and integration of GCMS data. Peak areas of all isotopologues for a molecular ion of each metabolite were used for mass isotopomer distribution analysis using a custom algorithm developed at VAI. Briefly, the atomic composition of the TBDMS-derivatized metabolite fragments (M-57) was determined, and matrices correcting for natural contribution of isotopomer enrichment were generated for each metabolite. After correction for natural abundance, a comparison was made between non-labeled metabolite abundances ($^{12}$C) and metabolite abundances which were synthesized from the $^{13}$C tracer. Metabolite abundance was expressed relative to the internal standard (D-myristic acid) and normalized to cell number.

For liquid chromatography coupled to mass spectrometry (LC-MS) metabolites were analyzed for relative abundance by high-resolution accurate mass detection on two QExactive Orbitrap mass spectrometers (Thermo Fisher Scientific) coupled to Thermo Vanquish liquid chromatography systems. Separate instruments were used for negative and positive mode analysis. For negative mode analysis, an Acquity T3 HSS (1.8 µm, 2.1 mm × 150 mm) column (Waters, Eschborn, Germany) was used for chromatographic separation and the elution gradient was carried out with a binary solvent system. Solvent A consisted of 3% methanol, 10 mM tributylamine, and 15 mM acetic acid in water (pH 5.0±0.05) and solvent B was 100% methanol. A constant flow rate of 200 µl/min was maintained and the linear gradient employed was as follows: 0–2.5 min 100% A, 2.5–5 min increase from 0% to 20% B, 5–7.5 min maintain 80% A and 20% B, 7.5–13 min increase from 20% to 55% B, 13–15.5 min increase from 55% to 95% B, 15.5–18.5 min maintain 5% A and 95% B, 18.5–19 min decrease from 95% to 0% B, followed by 6 min of re-equilibration at 100% A. The heater temperature was set to 400°C and ion spray voltage was set to 2.75 kV. The column temperature was maintained at 25°C and sample volumes of 10 µl were injected. A 22 min full-scan method was used to acquire data with *m/z* scan range from 80 to 1200 and resolution of 70,000. The automatic gain control (AGC) target was set at 1e6 and the maximum injection time was 500 ms. For positive mode analysis, an Atlantis T3 (3 µm, 2.1 mm ID×150 mm) column (Waters) was used and the elution gradient was carried out with a binary solvent system solvent A consisted of 0.1% acetic acid and 0.025% heptafluorobutyric acid in water and solvent B was 100% acetonitrile. A constant flow rate of 400 µl/min$^{-1}$ was maintained and the linear gradient employed was as follows: 0–4 min increase from 0% to 30% B, 4–6 min from 30% to 35% B, 6–6.1 min from 35% to 100% B and hold at 100% B for 5 min, followed by 5 min of re-equilibration. The heater temperature was set to 300°C and the ion spray voltage was set to 3.5 kV. The column temperature was maintained at 25°C and sample volumes of 10 µl were injected. An 11 min full-scan method was used to acquire data with *m/z* scan range from 70 to 700 and resolution of 70,000. The AGC target was set at 1e6 and the maximum injection time was 250 ms. Instrument control and acquisition was carried out by Xcalibur 2.2 software (Thermo Fisher Scientific). Full-scan LC-MS data were analyzed in Compound Discoverer (v.3.2, Thermo Scientific). Compounds were identified by chromatography specific retention time of external standards and MS2 spectral matching using the mzCloud database (Thermo Scientific).

## Quantification and statistical analysis

Two-way ANOVA was used to determine the effect of the genotype and treatment with the Prism 7.01 software (GraphPad Software). For repeated measures analysis, ANOVA was used when values over different times were analyzed. When only two groups were analyzed, statistical significance was determined by an unpaired Student's t-test. A value of $p < 0.05$ was considered statistically significant.

All data are shown as mean ± SEM unless stated otherwise. We did not include additional statistical tests for data distributions.

## Acknowledgements

We thank Karsten Hiller, Thekla Cordes, Heidi Lempradl, Connie Krawczyk, and Mylène Tajan for critical input and theoretical discussions. We thank Alexis Bergsma for technical support. We are indebted to the MPI-IE facilities and the VAI Vivari and facilities including the metabolomics, imaging, and genomics cores. This work was supported by funding from the MPG, the VAI, the European Union's Horizon 2020 research and innovation program under the Marie Skłodowska-Curie grant agreement no. 675610, the NeuroMac CRC/TRR167, and the Marie Skłodowska-Curie Postdoctoral Fellowship (EPOC – 707123). The team is supported by National Institutes of Health awards R21HG011964 and 1R01HG012444.

## Additional information

### Funding

| Funder | Grant reference number | Author |
| --- | --- | --- |
| Marie Sklodowska-Curie Actions | 10.3030/675610 | John Andrew Pospisilik |
| Marie Sklodowska-Curie Actions | 10.3030/707123 | Anne Drougard |
| Deutsche Forschungsgemeinschaft | NeuroMac - CRC/TRR167 | Marco Prinz John Andrew Pospisilik |
| National Institutes of Health | R21HG011964 | John Andrew Pospisilik |
| National Institutes of Health | 1R01HG012444 | John Andrew Pospisilik |

The funders had no role in study design, data collection and interpretation, or the decision to submit the work for publication.

### Author contributions

Anne Drougard, Conceptualization, Data curation, Formal analysis, Supervision, Funding acquisition, Validation, Investigation, Visualization, Methodology, Writing - original draft, Project administration, Writing - review and editing; Eric H Ma, Conceptualization, Formal analysis; Vanessa Wegert, Naman Vatsa, Judith Schaf, Klaus Gossens, Josephine Völker, Shengru Pang, Erez Dror, Francesca Giacona, Formal analysis; Ryan Sheldon, Sagar Sagar, Data curation, Formal analysis, Methodology; Ilaria Panzeri, Conceptualization, Writing - original draft; Stefanos Apostle, Software, Formal analysis; Luca Fagnocchi, Conceptualization, Data curation, Software, Writing - original draft; Anna Bremser, Conceptualization, Investigation, Methodology; Michael X Henderson, Methodology; Marco Prinz, Resources, Supervision, Writing - original draft; Russell G Jones, Resources, Methodology; John Andrew Pospisilik, Conceptualization, Resources, Funding acquisition, Investigation, Writing - original draft, Project administration, Writing - review and editing

### Author ORCIDs

Anne Drougard (ID) https://orcid.org/0000-0003-3029-6714
Naman Vatsa (ID) https://orcid.org/0000-0003-4683-3086
Luca Fagnocchi (ID) http://orcid.org/0000-0002-9551-5474
Judith Schaf (ID) https://orcid.org/0000-0002-8638-7490
John Andrew Pospisilik (ID) https://orcid.org/0000-0002-9745-0977

### Ethics

This study was performed in strict accordance with the recommendations in the Guide for the Care and Use of Laboratory Animals of the National Institutes of Health. All of the animals were handled according to approved institutional animal care and use committee (IACUC) protocol #22-07-027.

Reviewer #1 (Public review): https://doi.org/10.7554/eLife.87120.3.sa1
Reviewer #2 (Public review): https://doi.org/10.7554/eLife.87120.3.sa2
Reviewer #3 (Public review): https://doi.org/10.7554/eLife.87120.3.sa3
Author response https://doi.org/10.7554/eLife.87120.3.sa4

## Additional files

### Supplementary files

• MDAR checklist

### Data availability

scRNA sequencing data have been deposited in GEO under accession code GSE217045.

The following dataset was generated:

| Author(s) | Year | Dataset title | Dataset URL | Database and Identifier |
|---|---|---|---|---|
| Drougard A, Fagnocchi L, Pospisilik JA | 2024 | An acute microglial metabolic response controls metabolism and improves memory | http://www.ncbi.nlm.nih.gov/geo/query/acc.cgi?acc=GSE217045 | NCBI Gene Expression Omnibus, GSE217045 |

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
