## [Editor Report · eLife Assessment]

This **important** study demonstrates a link between an acute high fat diet, microglial metabolism and improved higher cognitive function. The evidence supporting the proposed mechanism in vivo is incomplete at this stage due to non-trivial technical limitations but the authors provide **convincing** in vitro metabolic characterization of primary microglia cultures to support the model. This work will be of interest to a broad audience in the field of neuroscience, metabolism, and immunology.

---

## [Referee Report · Reviewer #1 (Public review)]

In this study, Drougard et al. examined the consequences of an acute high fat diet (HFD) on microglia in mice. 3-day HFD influenced the regulation of systemic glucose homeostasis in a microglia-dependent and independent manner, as determined using microglial depletion with PLX5622. 3-day HFD increased microglial membrane potential and the levels of palmitate and stearate in cerebrospinal fluid in vivo. Using confocal imaging, respirometry and stable isotope-assisted tracing in primary microglial cultures, the authors suggest an increase in mitochondrial fission and metabolic remodelling occurs when exposed to palmitate, which increases the release of glutamate, succinate and itaconate that may alter neuronal metabolism. This acute microglial metabolic response following acute HFD is subsequently linked to improved higher cognitive function (learning and memory) in a microglia and DRP1-dependent manner.

Strengths:

Overall, this study is interesting and novel in linking acute high fat diet to changes in microglia and improved learning and memory in mice. The role for microglia and DRP1 in regulating glucose homeostasis and memory in vivo appears to be supported by the data. Palmitate (which is elevated in the CSF following acute HFD) is clearly used as a fuel by primary microglia ex vivo as determined using U-13C-plamitate tracing and metabolomics.

Weaknesses:

The authors suggest that utilisation of palmitate by microglia following HFD is the driver of the acute metabolic changes and that the release of microglial-derived lactate, succinate, glutamate and itaconate are causally linked to improvements in learning and memory. A weakness is that the authors provide no mechanistic link between beta-oxidation of palmitate (or other fatty acids) in microglia in vivo and the observed systemic metabolic and memory phenotypes. However, this reviewer acknowledges the technical difficulties of providing this evidence and approaches, such as microglia-specific deletion of CPT1a, will be an exciting avenue of research to explore for a subsequent study.

---

## [Referee Report · Reviewer #2 (Public review)]

The study by Drougard et al. aimed to answer a critical question on how high-fat diets trigger metabolic issues like obesity and diabetes. Their study revealed that an acute response by microglial cells in the brain to high-fat intake surprisingly benefits metabolism and cognitive function by rapidly metabolizing harmful fatty acids into alternative energy substrates like lactate and itaconate. Thus, short-term HFD intake seems to prompt a distinct beneficial response, suggesting a need for further exploration into the transition from acute to chronic effects.

---

## [Referee Report · Reviewer #3 (Public review)]

Drougard et al. explore microglial detection of a switch to high-fat diet and a subsequent metabolic response that benefits memory. The findings are both surprising and novel in the context of acute high-fat intake, with convincing evidence of increased CSF palmitate after 3 days of HFD. While the authors demonstrate compelling signs of microglial activation in multiple brain regions and unique metabolite release in tracing studies, they should address the following areas.

Major Points:

(1) It appears that the authors perform key metabolic assays in vitro/ex vivo using primary microglia from either neonatal or adult mice, which should be more clearly delineated especially for the 13C-palmitate tracing. In the case of experiments using primary microglia derived from mixed glial cultures stimulated with M-CSF, this system relies on neonatal mice. This is understandable given the greater potential yield from neonatal mice, but the metabolic state and energetic demands of neonatal and adult microglia differ as their functional roles change across the lifespan. The authors should either show that the metabolic pathways they implicate in neonatal microglia are also representative of adult microglia or perform additional experiments using microglia pooled from adult mice, especially because they link metabolites derived from neonatal microglia (presumably not under the effects of acute HFD) to improved performance in behavioral assays that utilize adult mice.

(2) The authors demonstrate that 3 days of HFD increases circulating palmitate by CSF metabolomics and that microglia can readily metabolize palmitate, but the causal link between palmitate metabolism specifically by microglia and improved performance in behavioral paradigms remains unclear. A previous body of research, alluded to by the authors, suggests that astrocyte shuttling of lactate to neurons improves long-term and spatial memory. The authors should account for palmitate that also could be derived from astrocyte secretion into CSF, and the relative contribution compared to microglia-derived palmitate. Specifically, although microglia can metabolize the palmitate in circulation, there is no direct evidence that the palmitate from the HFD is directly shuttled to microglia and not, for example, to astrocytes (which also express CX3CR1). Thus, the Barnes Maze results could be attributed to multiple cell types. Furthermore, the evidence provided in Figure 5J is insufficient to claim a microglia-dependent mechanism without showing data from mice on HFD with and without microglia depletion (analogous to the third and fourth bars in panel K).

(3) Given the emphasis on improved cognitive function, there is minimal discussion of the actual behavioral outcomes in both the results and discussion sections. The data that HFD-treated animals outperform controls should be presented in more detail both in the figure and in the text. For example, data from all days/trials of the Barnes Maze should be shown, including the day(s) HFD mice outperform controls. Furthermore, the authors should either cite additional literature or provide experimental evidence supporting the notion that microglia release of TCA-associated substrates into the extracellular milieu after HFD specifically benefits neuronal function cellularly or regionally in the brain, which could translate to improved performance in classical behavioral paradigms. The single reference included is a bit obscure, given the study found that increased lactate enhances fear memory which is a neural circuit not studied in the current manuscript. Are there no additional studies on more relevant metabolites (e.g., itaconate, succinate)?

---

## [Author Response]

The following is the authors’ response to the previous reviews.

**Public Reviews:**

**Reviewer #1 (Public Review):**
In this study, Drougard et al. examined the consequences of an acute high fat diet (HFD) on microglia in mice. 3-day HFD influenced the regulation of systemic glucose homeostasis in a microglia-dependent and independent manner, as determined using microglial depletion with PLX5622. 3-day HFD increased microglial membrane potential and the levels of palmitate and stearate in cerebrospinal fluid in vivo. Using confocal imaging, respirometry and stable isotope-assisted tracing in primary microglial cultures, the authors suggest an increase in mitochondrial fission and metabolic remodeling occurs when exposed to palmitate, which increases the release of glutamate, succinate and itaconate that may alter neuronal metabolism. This acute microglial metabolic response following acute HFD is subsequently linked to improved higher cognitive function (learning and memory) in a microglia and DRP1-dependent manner.Strengths:Overall, this study is interesting and novel in linking acute high fat diet to changes in microglia and improved learning and memory in mice. The role for microglia and DRP1 in regulating glucose homeostasis and memory in vivo appears to be supported by the data.Weaknesses:The authors suggest that utilization of palmitate by microglia following HFD is the driver of the acute metabolic changes and that the release of microglial-derived lactate, succinate, glutamate and itaconate are causally linked to improvements in learning and memory. A major weakness is that the authors provide no mechanistic link between beta-oxidation of palmitate (or other fatty acids) in microglia and the observed systemic metabolic and memory phenotypes in vivo. Pharmacological inhibition of CPT1a could be considered or CPT1a-deficient microglia.

We thank Reviewer #1 for their time, effort and the critique. Indeed, we suggest that palmitate drives the aMMR response and associated improvements in learning and memory. In response to acute HFD we observe (1) increased in palmitate in CSF; (2) impaired mitochondrial ETC activity in primary microglia (within 12 hours of HFD); and (3) improved learning and memory. The greatest barrier to proving how acute palmitate uptake in microglia improves learning and memory in vivo is the protracted methodology required for microglial isolation and purification. The timeframes and relatively harsh digestion protocols required are currently incompatible with metabolomic tracing and well beyond those required for most cell-types used for metabolomic investigation. We have tested and failed to obtain reproducible data across numerous in vivo protocols and finally settled on in vitro 13C palmitate treated neonatal microglia as the best current option. Primary neonatal microglia are accepted as one of the current best culture models by the microglial community (Valdercaos cell report 2014, Kim Cell Metab 2019). Using neonatal microglia we demonstrate that 13Cpalmitate label is processed to palmitoylcarnitine (Fig 4C) and acetylcarnitine (Fig 4D) indicating that microglial fatty acid metabolism acts via the canonical CPT1/CPT2 pathway. These experiments highlight that microglia process palmitate via beta oxidation generating acetyl coA and engaging the TCA cycle (Fig 4G-I).

We now acknowledge these technical limitations more clearly and highlight their impact on any conclusions regarding adult microglia in vivo:

Results “Microglia take up and metabolize free fatty acids”;

“Due in part to the long isolation times required to generate pure primary adult microglia, metabolite tracing experiments on primary adult microglia are not currently feasible. We therefore chose primary murine neonatal microglia as our model of choice for more mechanistic experiments (Valdercaos, Cell Report 2014)”

And,

Discussion:

“We propose that aMMR could result from direct uptake, processing, and release of fatty acid derived carbons, and demonstrate that microglia are capable of metabolizing fatty acids towards diverse intracellular and extracellular pools.”

While acute ICV injection a CPT1a blocker would be of potential interest, the caveats associated with CPT1a inhibition in other cell-types (neurons, astrocytes, etc) and with targeting the appropriate brain region (currently unknown) currently preclude the effective use of this approach for to generate clear additional mechanistic insights. Similarly, given the time and resources required to generate, validate, optimize and experiment on a clean model of in vivo adult microglia-specific CPT1a knockout, this approach was deemed beyond the scope of this study. That said, the critique is important, and it should comprise a follow-up project.

Comment: Another major weakness is that the authors also suggest that 3-day HFD microglial response (increase membrane potential) is likely driven by palmitate-induced increases in itaconate feedforward inhibition of complex II/SDH. Whilst this is an interesting hypothesis, the in vitro metabolic characterization is not entirely convincing.

The reviewer is correct, we suggest that our data is consistent with a model where a palmitate-induced increase in itaconate inhibits complex II/SDH. While our findings do not comprise mechanistic proof, the hypothesis is supported by our Seahorse studies (Fig 2E) highlighting that a combined Palmitate + Succinate stimulation does not increase OCR beyond that of Palmitate alone; by primary microglial cell experiments highlighting that 3d-HFD treated adult primary microglia are refractory to succinate-induced mitochondrial membrane depolarization (Fig 2F); and by the identification of increased palmitate induced itaconate production/release in cultured primary neonatal microglia (Fig 4H). The data are consistent with an inhibition of complex II/ SDH and with increased itaconate secretion. They are also consistent with literature on more easily accessible myeloid lineages (Lampropoulou V, Cell Metab 2016).

Comment: The authors suggest that acute palmitate appears to rapidly compromise or saturate complex II activity. Succinate is a membrane impermeable dicarboxylate. It can enter cells via MCT transporters at acidic pH. It is not clear that (I) Succinate is taken up into microglia, (II) If the succinate used was pH neutral sodium succinate or succinic acid, and (III) If the observed changes are due to succinate oxidation, changes in pH or activation of the succinate receptor SUCNR1 on microglia. In the absence of these succinate treatments, there are no alterations in mitochondrial respiration or membrane potential following palmitate treatment, which does not support this hypothesis.

We thank Reviewer #1 for highlighting a lack of information in the material and methods. We have updated them accordingly as follows:

“For the electron transport chain experiments (ETC), the experiment was based on the Salabei et al. The cell suspension was incubated with the mitochondrial probe Tetramethylrhodamine TMRM (10mM; Abcam, Cat# ab228569) and fluorescent glucose analog 2-NBDG (Abcam, Cat# 235976) for 30min at 37degrees before FACS acquisition. For challenging the ETC, the cell pellet was resuspended in 500ul of warm MAS buffer solution + 1nM Plasma Membrane Permeabilizer (Agilent Seahorse XF PMP) in order to permeabilize the cells. Microglial cells were gated from CD45low-CD11b+ cells followed by singlet after forward and side scatter pattern. They were incubated each 90 seconds by the following drugs: 0,5ul of 100uM Rotenone (Sigma), 2ul of 2.5M Succinate adjusted to ph 7.4 with NaOH (succinic acid, Sigma) and 0.5ul of 1mM Antimycin (Sigma). Cytometry was performed on Fortessa (BD Bioscience) and analyzed with FlowJo v10 (Treestar).”

Following the updated protocol, we hope we highlighted that the succinate (solution of succinic acid ph 7.4) is reaching directly the ETC since the microglial cells have been permeabilized by the Plasma Membrane Permeabilizer (Agilent Seahorse XF PMP).

Comment: Intracellular itaconate measurements and quantification are lacking and IRG1 expression is not assessed. There also appears to be more labelled itaconate in neuronal cultures from control (BSA) microglia conditioned media, which is not discussed. What is the total level of itaconate in neurons from these conditioned media experiments? No evidence is provided that the in vivo response is dependent on IRG1, the mitochondrial enzyme responsible for itaconate synthesis, or itaconate. To causally link IRG1/itaconate, IRG1-deficient mice could be used in future work.

We appreciate the interest, the exciting question, and the suggested future experiment. Indeed, our results suggest a difference in metabolite release between the BSA treated-microglia and palmitate treated-microglia and their impact on neurons comprises a prime question for future work. We have highlighted this in the discussion as well as adding a comment regarding relative levels of labelled itaconate as follows:

Results; Acute HFD induces widespread MMR and rapid modulation (…) memory

“As a control for the direct uptake of 13C-glucose, we treated parallel neuronal cultures with the same fresh 13C-glucose tracing media originally added to the microglia. Intriguingly, and consistent with literature documenting poor direct glucose utilization by neurons [29], we found substantial m+3 lactate (as well as other metabolites) in neurons treated with microglial conditioned media, and at levels that far exceeded labelling triggered by glucose tracer alone (Fig 5A, middle column vs left column)(Suppl Fig S5B). The data indicate higher uptake of citrate and itaconate from the control microglia-conditioned media, further supporting the hypothesis that neuronal metabolism is reproducibly impacted by palmitate-triggered changes in microglial products. These data demonstrate that palmitate metabolism by microglia modulates neuronal carbon substrate use in vitro, and, they highlight the relative importance of this process compared to uptake of pure glucose. The data identify a candidate mechanism by which aMMR may alter neuronal function in vivo.”

Comment: While microglial DRP1 is causally implicated the role of palmitate is not convincing. Mitochondrial morphology changes are subtle including TOMM20 and DRP1 staining and co-localization - additional supporting data should be provided. Electron microscopy of mitochondrial structure would provide more detailed insight to morphology changes. Western blot of fission-associated proteins Drp1, phospho-Drp1 (S616), MFF and MiD49/51. Higher magnification and quality confocal imaging of DRP1/TOMM20. Drp1 recruitment to mitochondrial membranes can be assessed using subcellular fractionation.

We appreciate the reviewer’s comment. Previous work by others, already cited elsewhere in our manuscript

(PMCID: PMC7251564), has clearly demonstrated increased mitochondrial fragmentation and

phosphorylated DRP1 in 3d HFD animals. This very specific result can therefore be considered confirmatory / validating of existing literature, and important for inclusion of DRP1 in our overall model. We have made sure to better highlight this important literature accordingly:

Results; A rapid Microglial Mitochondria response to high fat diet

“Consistent with the in vivo observations above, in vitro palmitate exposure decreased microglial mitochondrial length within 24 hours, indicating that fatty acid exposure itself is sufficient to trigger mitochondrial fission in a cell autonomous manner (Fig 2G upper panels). This result also confirms observations by Kim et al. who observed mitochondrial fission and DRP1 phosphorylation upon 3d-HFD treated mice [Kim JD et al, Microglial UCP2 mediates Inflammation and Obesity induced by High Fat feeding, Cell Metab 2019].”

Comment: No characterization of primary microglia from DRP1-knockout mice is performed with palmitate treatment. Authors demonstrate an increase in both stearate and palmitate in CSF following 3day HFD. Only palmitate was tested in the regulation of microglial responses, but it may be more informative to test stearate and palmitate combined.

Testing stearate and palmitate combined is an interesting experiment for mimicking the global effect of HFD which is highly enriched with these two satured fatty acids, and then, more informative. In vitro stimulation of microglia model cells has been previously published by Valdearcos and al. (Cell Reports 2014) who studied the effect of a mix of stearate and palmitate on the Mediobasal Hypothalamus inflammation. Here, we build on their important findings by demonstrating that these 2 compounds are actually found in the CSF of 3d-HFD mice. Studies from other labs have also shown the presence of stearate and palmitate in the CSF of chronically obese and diabetic patients which highlights the importance of these findings (Melo HM et al. cell report 2020). While a systematic dissection of the roles of HFD-regulated CSF metabolites (including direct (diet containing) and indirect (secondary) is beyond the scope of this study), this point is important, not least because it highlights less well-studied metabolites and the potential of possible combinatorial interactions. We have highlighted this idea in the results as follows:

Results; A rapid Microglial Mitochondria response to high fat diet

“To test whether these observed fatty acid changes in the CSF might directly trigger aMMR, we switched to an in vitro primary neonatal microglia model and examined the effects of the more abundant of these, palmitate (Fig S2A-B).”

and, in the discussion as follows:

“Studies have identified stearate and palmitate in the CSF of patients with chronic obesity and with diabetes, reports that highlight the importance of these findings (Melo HM et al. cell report 2020). While a systematic dissection of the roles of HFD-regulated CSF metabolites (including direct (diet containing) and indirect (secondary)) is beyond the scope of this study, they represent priority areas for future investigation, particularly given the wide-range of fatty-acid specific biological effects in the literature, and the potential for combinatorial interactions.”

**Reviewer #1 (Recommendations For The Authors):**
Congratulations on this interesting and novel work. Please see public review for details on potential experiments. While I would not expect all the experiments to be performed for this current study, it’s important to not overstate what the data is showing. For example, there is no causal link between palmitate oxidation in microglia or released metabolites (itaconate etc) from microglia in the effect on systemic glucose metabolism or memory. To make such claims more supporting data would be required.

We thank Reviewer #1 for their highly constructive critique_._

**Reviewer #2 (Public Review):**
The study "A rapid microglial metabolic response controls metabolism and improves memory" by Drougard et al. provides evidence that short-term HFD has a beneficial effect on spatial and learning memory through microglial metabolic reprogramming. The manuscript is well-written and the statistics were properly performed with all the data. However, there are concerns regarding the interpretation of the data, particularly the gap between the in vivo observations and the in vitro mechanistic studies.In the PLX-5622 microglial depletion study, it is unclear what happened to the body weight, food intake, and day-night behavior of these mice compared to the vehicle control mice. It is important to address the innate immunity-dependent physiology affected by a long period of microglial depletion in the brain (also macrophages in the periphery). Furthermore, it would be beneficial to validate the images presented in Fig.1F by providing iba1 staining in chow diet-fed mice with or without PLX-5622 for 7-10 days. Additionally, high-quality images, with equal DAPI staining and comparable anatomical level, should be provided in both chow diet-fed mice and HFD-fed mice with or without PLX-5622 in the same region of hypothalamus or hippocampus. These are critical evidences for this project, and it is suggested that the authors provide more data on the general physiology of these mice, at least regarding body weight and food intake.

We are grateful to Reviewer #2 for their constructive comments and for their time and effort; and for highlighting the lack of experimental details regarding the PLX-5622 microglial depletion study. We followed the protocol established in Feng et al JCI 2017. No adverse effects on body weight, food intake and day-night behavior have been described in this study as well as in other studies for longer treatment (Sonia George et al Molecular Neurodegeneration 2019). We didn’t observe any differences in body weight and the food intake within or between groups, upon PLX administration. These data have been included as new Supplementary Fig 6 A-B.

The material and method was updated as follows:

“Animals were administered PLX5622-containing diet for 7-9 days without observable impact on the body weight or food intake (Fig S6A-B), using protocols adopted from [Feng et al JCi 2017, Sonia George et al Molecular Neurodegeneration 2019].”

Comment: It is also unclear whether the microglia shown in Fig.3A were isolated from mice 4 weeks after Tamoxifen injection. It is suggested that the authors provide more evidence, such as additional images or primary microglia culture, to demonstrate that the mitochondria had more fusion upon drp1 KO. It is recommended to use mito-tracker green/red to stain live microglia and provide good resolution images.

We thank Reviewer #2 for pointing out the lack of detailed information about Fig 3A. Microglial cells were indeed isolated from mice after the tamoxifen injection for highlighting the deletion. We updated the Material and methods with the text below;

“For the colocalization experiment, microglia were isolated from 10 to 12-week old drp1ko mice and their littermate controls, immediately fixed in PFA and stained with DRP1 (diluted 1:50 Cell signaling; Cat#8570) and tomm20 antibodies (diluted 1:1000, SantaCruz; Cat#sc177615).”

This experiment was performed as an additional control of the drp1 deletion from our knockout-mice. For this experiment we used Tomm20 since the microglia cells weren’t live after the addition of PFA.

Comment: Regarding the data presented in Fig.5A, it is suggested that the authors profile the metabolomics of the microglial conditioned media (and provide the methods on how this conditioned media was collected) to determine whether there was already abundant lactate in the media. Any glucose-derived metabolites, e.g. lactate, are probably more preferred by neurons as energy substrates than glucose, especially in embryonic neurons (which are ready to use lactate in newborn brain).

With regards to Fig 5A, metabolomics of microglia conditioned media are provided as Fig 5A, Supp Figure 5Band we provided a supplementary table 2.

We thank Reviewer #2 for noting the lapse of technical detail. We updated the Material and methods with the following:

“For conditioned media experiments, microglial cells were incubated with DMEM (Gibco) without lactate completed with BSA-conjugated palmitate or Control BSA. Conditioned media was collected after the incubation, centrifuged 15min at 300g (4oC) and the supernatant transferred and frozen in a fresh tube avoiding the cells and debris pellet. Sample were immediately snap frozen or use for the neurons incubation.”

Any glucose-derived metabolites, e.g. lactate, are more preferred by neurons as energy substrates than glucose as described first in the literature by Prof. Pellerin and Prof. Magistretti via the astrocyte-neuron cooperation (PNAS 1994). Since their discovery, lactate has been explored and is well known as a key signaling molecule (Magistretti PJ Nat Rev Neurosciences 2018). We explored the role of lactate released from the microglia, and we demonstrated that it is taken up by neurons independently of any microglial pretreatment. This experiment highlights microglia as another lactate provider for the neurons (Fig 4N and Fig 5A).

Comment: Finally, it is important to address whether PLX-5622 affects learning and spatial memory in chow diet-fed animals. Following the findings shown in Fig 5J and 5K, the authors should confirm these by any morphological studies on synapse, e.g. by synaptophysin staining or ultrastructure EM study in the area shown in Fig 5I.

We appreciate the comment and question. We performed the controls and included them now as Fig 5J and Fig S5 E-F-G. We do not observe any adverse effects of PLX5622 on learning and spatial memory in normal chow-fed animals.

While we were unable to study the synapses as requested, it is important to note that no changes are expected given publications from other labs using the same protocol (Feng x JCI 2017 ,Spangenberg E Nat Com 2019), or longer PLX5622 treatment (Niiyama T eNeuro 2023, Witcher KG J neurosciences 2021), all four of which did not find morphological differences at synapses.

**Reviewer #2 (Recommendations For The Authors):**
The authors should provide more evidence that palmitate is derived from HFD to prove that it mediates the HFD effects on the microglial mitochondria response. This could be done by adding 13C-palmitate into the HFD and performing metabolomics in isolated microglia from control mice (and Drp1-MG-KO mice, if possible).

We thank the Reviewer #2 for the enthusiastic revision. Unfortunately, we were unable to attempt this final suggested experiment. We have adjusted our wording accordingly and appreciate the reviewer’s understanding.

**Reviewer #3 (Public Review):**
Drougard et al. explore microglial detection of a switch to high-fat diet and a subsequent metabolic response that benefits memory. The findings are both surprising and novel in the context of acute highfat intake, with convincing evidence of increased CSF palmitate after 3 days of HFD. While the authors demonstrate compelling signs of microglial activation in multiple brain regions and unique metabolite release in tracing studies, they should address the following areas prior to acceptance of this manuscript.Major Points:(1) It appears that the authors perform key metabolic assays in vitro/ex vivo using primary microglia from either neonatal or adult mice, which should be more clearly delineated especially for the 13C-palmitate tracing. In the case of experiments using primary microglia derived from mixed glial cultures stimulated with M-CSF, this system relies on neonatal mice. This is understandable given the greater potential yield from neonatal mice, but the metabolic state and energetic demands of neonatal and adult microglia differ as their functional roles change across the lifespan. The authors should either show that the metabolic pathways they implicate in neonatal microglia are also representative of adult microglia or perform additional experiments using microglia pooled from adult mice, especially because they link metabolites derived from neonatal microglia (presumably not under the effects of acute HFD) to improved performance in behavioral assays that utilize adult mice.

We thank Reviewer #3 for their constructive critique and encouraging words. As indicated, the 13C-palmitate experiments were performed with primary microglia derived from mixed glial cultures stimulated with M-CSF and we demonstrated our primary cultures were almost pure by the supplementary experiments (supp Fig2A and B). Additional minor details in these contexts have been added to the Material and Methods.

The experiments focusing on the mitochondrial ETC were performed on sorted microglia from adult mice and parallels demonstrated with the neonatal cultures (the primary model for metabolic tracing). Compromised complex II activity under conditions of acute HFD/palmitate stimulation for instance were shown in both systems. Unfortunately, despite best-efforts, attempts to run 13C-palmitate tracing experiments on primary adult microglia failed, attributable in large part to the long (~4 hour) and harsh microglial extraction and sorting process. These experiments will require substantial follow-up efforts including the establishment and validation ideally of an adult microglia-neuron co-culture model that faithfully recapitulates most aspects of in vivo metabolic cross-talk. This noble aim is beyond the scope of this study. We have made sure to temper the conclusions made in the manuscript and to not overstate the impact and interpretation of the in vitro work including updating the following sentences.

Results “Microglia take up and metabolize free fatty acids”;

“Due in part to the long isolation times required to generate pure primary adult microglia, metabolite tracing experiments on primary adult microglia are not currently feasible. We therefore chose primary murine neonatal microglia as our model of choice for more mechanistic experiments (Valdercaos cell Report 2014)”

and Discussion:

“We propose that aMMR could result from direct uptake, processing, and release of fatty acid derived carbons, and demonstrate that microglia are capable of metabolizing fatty acids towards diverse intracellular and extracellular pools.”

Comment: The authors demonstrate that 3 days of HFD increases circulating palmitate by CSF metabolomics and that microglia can readily metabolize palmitate, but the causal link between palmitate metabolism specifically by microglia and improved performance in behavioral paradigms remains unclear. A previous body of research, alluded to by the authors, suggests that astrocyte shuttling of lactate to neurons improves long-term and spatial memory. The authors should account for palmitate that also could be derived from astrocyte secretion into CSF, and the relative contribution compared to microglia-derived palmitate. Specifically, although microglia can metabolize the palmitate in circulation, there is no direct evidence that the palmitate from the HFD is directly shuttled to microglia and not, for example, to astrocytes (which also express CX3CR1).

We appreciate the comment. Indeed, this issue highlights one of the greatest challenges for efforts aimed at tracing (beyond doubt) that a single minor cell population contributes towards metabolic cross-talk in vivo. Our experiments show: increased CSF palmitate levels within one feeding cycle of HFD; rapidly induced microglial metabolic activation (characterized by increased mitochondrial membrane potential and impaired complex II activity); and that microglia mount a comparable mitochondrial activation profile in vitro when exposed to palmitate. They show in vitro using neonatal microglia that microglia take up and metabolize palmitate; that they release metabolites with neuro-modulatory potential; that neurons take these metabolites up and modulate their function differentially when exposed to control vs palmitate-treated microglia-conditioned media (in the absence of astrocytes). The experiments show through acute PLX-induced elimination of microglia, however crude, that this compartment impacts the acute HFD response, and using conditional deletion, that full DRP1 expression is required CX3CR1-CreERT2 targeted cells (primarily microglia deleting; Zhao et al 2019). While these experiments cannot rule out a contribution of astrocytes to the observations in vivo, comparable experiments rarely can and we cannot rationalize why microglia should not have equal access to CSF palmitate for uptake or to neurons for substrate provisioning. We now better highlight this important issue, and temper our conclusions accordingly:

“Tanycytes and astrocytes have both been documented to release select metabolites into the extracellular environment [33, 34]. While suggestive, the experiments highlighted here do not rule out a contribution of these or cell types in coupling acute HFD intake to memory and learning.”

Comment: Thus, the Barnes Maze results could be attributed to multiple cell types. Furthermore, the evidence provided in Figure 5J is insufficient to claim a microglia-dependent mechanism without showing data from mice on HFD with and without microglia depletion (analogous to the third and fourth bars in panel K).

Agreed. We appreciate the comment. We have now added the requested HFD condition to Figure 5J. The data support our previous interpretation of the data.

Comment: Given the emphasis on improved cognitive function, there is minimal discussion of the actual behavioral outcomes in both the results and discussion sections. The data that HFD-treated animals outperform controls should be presented in more detail both in the figure and in the text. For example, data from all days/trials of the Barnes Maze should be shown, including the day(s) HFD mice outperform controls. Furthermore, the authors should either cite additional literature or provide experimental evidence supporting the notion that microglia release of TCA-associated substrates into the extracellular milieu after HFD specifically benefits neuronal function cellularly or regionally in the brain, which could translate to improved performance in classical behavioral paradigms. The single reference included is a bit obscure, given the study found that increased lactate enhances fear memory which is a neural circuit not studied in the current manuscript. Are there no additional studies on more relevant metabolites (e.g., itaconate, succinate)?

We agree. We have now re-plotted the behavioral test to better highlight that the HFD-treated animals outperform controls, as requested (Fig S7 and S8). We also added the requested literature. While we cannot be sure our search captured all relevant studies, we find a relative paucity of studies that characterize CSF metabolite changes in the context of acute high fat feeding or that demonstrate the ability of CSF substrates to convincingly improve memory and learning in vivo at physiological levels. Indeed, while simple, we feel the findings are of substantial novelty and highlight an area for significant future research. We have tempered our conclusions throughout and added to the discussion as follows:

“Such substrate release could mediate the learning and memory effects that accompany aMMR; they are consistent with the data of other studies that have examined metabolite associations with learning and memory (itaconate [Morgunov IG, microorganisms 2020; Xiong J, Neuromolecular med 2023], succinate [Serra FT neurosciences letter 2022; Cline BH, BMC neurosciences 2012]).”

Minor Points:(1) In Figure 5J the latency to find the hole was noticeably higher (mean around 150s) than the latency in panel K (mean around 100s for controls, and 60s for Drp1MGWT on HFD). This suggests high variability between experiments using this modified version of the Barnes Maze, despite the authors assertion that a standard Barnes Maze was employed and the results were reproducible at multiple institutions. Why do Drp1MGWT mice on control diet find the escape hole significantly faster than WT mice on control diet in panel J? Given the emphasis on cognitive improvement following acute HFD as a novel finding, the authors should explain this discrepancy.

We appreciate this question and comment. Indeed, as the reviewer knows, behavioral tests including the Barnes test show variation with genetic background, and with environment and context (eg. age, caging density, litter size, behavioral state and more (Inglis A, Physiol Behavior 2019; Loos M Mamm Genome 2015; and unpublished observations)). We do not know the exact origin of the difference mentioned above but our best guess would be that it stems from either environmental differences that are ever present in vivaria (seasonal, mouse house room, cage-changing cycles, etc) and/or, differences between the background genetics (eg. presence of Cre transgene and linked genome, genetic drift) or precise experimental differences between the cohorts (eg. repeated tamoxifen-injection paradigm for the deletion group). All of our experiments were performed in parallel, with all relevant animal groups equally represented in every run, and,and used age- and sex-matched individuals from congenic strains. Wherever possible, controls and test animals were *littermates* to minimize within strain variance attributable to litter effects (litter size, maternal and paternal effects). Given our lab’s interest and focus on the mechanistic and developmental origins of variance heterogeneity, these differences are of high interest for future study.

Comment: The authors highlight in the graphical abstract and again in Figure 4A the formation of lipid droplets following palmitate exposure as evidence of that microglia can process fatty acids. They later suggest that a lack of substantial induction of lipid droplet accumulation suggests that microglia are metabolically wired to release carbon substrates to neighboring cells. Clarification as to the role of lipid droplet formation/accumulation in explaining the results would eliminate any possible confusion.

We modified the wording in the manuscript accordingly:

Results “Microglia take up and metabolize free fatty acids”;

“Based on BODIPY fluorescence, we found that primary microglia increase lipid droplet numbers within 24h of in vitro exposure to palmitate (200uM; Fig 4A), demonstrating a capacity to take up fatty acids.”

Comment: In many bar graphs showing relatively modest effects, it would be helpful to use symbols to also show the distribution of sample and animal replicates (especially behavioral paradigms).

Agreed. Indeed, the results are both modest and impressive given the nature of the intervention (simple change in dietary macronutrient composition). We have now re-plotted the results from the behavioral experiments, accordingly (Fig S7 and Fig S8).

**Reviewer #3 (Recommendations For The Authors):**
This is a good manuscript deserving of publication assuming some of the concerns posed above are addressed.

We thank Reviewer #3 again for their time, effort, and dedication, and for their objective review of the manuscript.